# Predict-then-Calibrate: A New Perspective of Robust Contextual LP

**Chunlin Sun**[*,1]   **Linyu Liu**[*,2]   **Xiaocheng Li**[3]

[1] Institute for Computational and Mathematical Engineering, Stanford University
[2] Department of Automation, Tsinghua University
[3] Imperial College Business School, Imperial College London

## Abstract

Contextual optimization, also known as predict-then-optimize or prescriptive analytics, considers an optimization problem with the presence of covariates (context or side information). The goal is to learn a prediction model (from the training data) that predicts the objective function from the covariates, and then in the test phase, solve the optimization problem with the covariates but without the observation of the objective function. In this paper, we consider a risk-sensitive version of the problem and propose a generic algorithm design paradigm called predict-then-calibrate. The idea is to first develop a prediction model without concern for the downstream risk profile or robustness guarantee, and then utilize calibration (or recalibration) methods to quantify the uncertainty of the prediction. While the existing methods suffer from either a restricted choice of the prediction model or strong assumptions on the underlying data, we show the disentangling of the prediction model and the calibration/uncertainty quantification has several advantages. First, it imposes no restriction on the prediction model and thus fully unleashes the potential of off-the-shelf machine learning methods. Second, the derivation of the risk and robustness guarantee can be made independent of the choice of the prediction model through a data-splitting idea. Third, our paradigm of predict-then-calibrate applies to both (risk-sensitive) robust and (risk-neutral) distributionally robust optimization (DRO) formulations. Theoretically, it gives new generalization bounds for the contextual LP problem and sheds light on the existing results of DRO for contextual LP. Numerical experiments further reinforce the advantage of the predict-then-calibrate paradigm in that an improvement on either the prediction model or the calibration model will lead to a better final performance.

## 1   Introduction

Contextual linear programming (LP) considers the linear programming problem with the presence of covariates, and it can be viewed as a prediction problem under a constrained optimization context where the output of the prediction model serves as the objective of a downstream LP. The goal is to develop a model (trained from past data) that prescribes a decision/solution for the downstream LP problem using directly the covariates but without observation of the objective function. A similar problem formulation was also studied as prescriptive analytics [Bertsimas and Kallus, 2020] and predict-then-optimize Elmachtoub and Grigas [2022]. The existing literature mainly develops two approaches for the problem: (i) end-to-end algorithms that map the covariates directly to a recommended decision for the LP or (ii) two-step algorithms that first predict the unobserved objective using the covariates and then solve the LP with the predicted objective. While the literature has been largely focused on the risk-neutral objective, our paper considers the risk-sensitive/robust

---

[*]Equal contribution.

37th Conference on Neural Information Processing Systems (NeurIPS 2023).

formulation of the problem. The robust formulation of the contextual LP can be viewed as both a risk-sensitive version of the risk-neutral objective in contextual LP and an adaptive/contextual version of the standard (context-free) robust optimization problem. In comparison with the robust optimization literature, the contextualized robust optimization problem allows a contextual and data-driven way to design the uncertainty set and ideally can utilize the contextual information to make less conservative decisions. The contextual/data-driven uncertainty set has been considered by several existing works on transportation (Guo et al. [2023]), portfolio management (Wang et al. [2022]), and healthcare (Gupta et al. [2020]). While these works focus on applications with special structures or distributions, our paper considers a generic setup and aims for a general solution to the problem.

In our paper, we make the following contribution:

First, we formulate an algorithm design paradigm of predict-then-calibrate and introduce the notion of uncertainty calibration/quantification to the problem of contextual optimization. The idea is to first develop a prediction model without worrying about the downstream robustness, and then tailor the uncertainty calibration methods to quantify the prediction uncertainty towards the robust objective.

Second, we develop two algorithms for robust contextual LP under the paradigm of predict-then-calibrate. The derivation of theoretical guarantees shows the advantage of the disentanglement of the prediction phase and the uncertainty calibration phase. Moreover, we quantify the advantage of a better prediction and/or a better uncertainty calibration both theoretically and empirically.

Third, we also draw a connection with the related problem of distributionally robust optimization. We utilize the same idea of predict-then-calibrate and derive a new generalization bound for the distributional robust version of contextual LP utilizing tools from the nonparametric regression.

In the following, we review the several streams of literature that are related to our work.

**Conditional robust optimization.** The problem studied in our paper can be viewed as a conditional version of the classic robust optimization problem. The robust optimization stems from the replacement of the risk-neutral expectation objective with a risk-sensitive robust objective such as Value-at-Risk(VaR)/quantile. The literature on robust optimization develops methods to approximate VaR by specifying the shape and size of an uncertainty set to guarantee that a VaR constraint would be satisfied Ghaoui et al. [2003], Chen et al. [2007], Natarajan et al. [2008]. Bertsimas et al. [2018] designs the uncertainty set with a finite-sample probabilistic guarantee on optimal solutions. Goerigk and Kurtz [2020] constructs the uncertainty set as the union of $K$ convex sets by clustering historical data. While all these works concern the context-free setting, the contextual (or conditional) setting, or the problem of robust contextual LP, considers the presence of covariates in robust optimization. Along this line, Ohmori [2021] proposes a volume-minimized ellipsoid covering $k$-nearest samples in the contextual space, leading to a nonlinear programming problem. Chenreddy et al. [2022] extends the idea in Goerigk and Kurtz [2020] and novelly adopts a deep learning approach to the problem which identifies which clusters the uncertain parameters belong to based on the covariates. Compared to the end-to-end formulations Ohmori [2021], Chenreddy et al. [2022], we do not impose restrictions on the prediction model and have a larger flexibility in choosing the uncertainty quantification method.

**Contextual LP/Predict-then-optimize.** Our work complements the existing literature on contextual LP by considering a risk-sensitive objective. The existing works Donti et al. [2017], Elmachtoub and Grigas [2022], Ho-Nguyen and Kılınç-Karzan [2022], Hu et al. [2022] mainly study the risk-neutral objective for the problem and aim to develop prediction methods that account for the downstream optimization tasks. To model the remaining uncertainty after prediction in optimization, Sen and Deng [2018] considers training a parameterized point-wise estimation model first, and then approximating the conditional distribution by imposing randomness to the learned parameters empirically. Ban et al. [2019] proposes to forecast the distribution of new product demand by adopting a linear regression model to fit the demands of similar products with covariates, and then using the empirical distribution of covariate-independent residuals to build a sample average approximation (SAA) model. Some recent works Kannan et al. [2020, 2022, 2021] further generalize it to a residuals-based SAA framework to include more machine learning methods and adopt distributionally robust optimization to mitigate the overfitting under limited data. A key difference between our work and this line of works is that we consider a risk-sensitive objective, while these works consider a risk-neutral objective. In this light, our framework of predict-then-calibrate is compatible with any prediction model developed along this line, and it tailors the prediction model for a robustness task.

## 2 Problem Setup

Consider a linear program (LP) that takes the following standard form

$$\text{LP}(c, A, b) := \min c^\top x, \tag{1}$$
$$\text{s.t. } Ax = b, \ x \geq 0,$$

where $c \in \mathbb{R}^n$, $A \in \mathbb{R}^{m \times n}$, and $b \in \mathbb{R}^m$ are the inputs of the LP. In addition, there is an available feature vector $z \in \mathbb{R}^d$ that encodes the covariates (side information) associated with the LP.

The tuple $(c, A, b, z)$ is drawn from an unknown probability distribution $\mathcal{P}$. The problem of contextual LP (predict-then-optimize, or prescriptive analytics) usually assumes the availability of a learning (training) dataset sampled from the distribution $\mathcal{P}$,

$$\mathcal{D}_l := \left\{ \left( c_t^{(l)}, A_t^{(l)}, b_t^{(l)}, z_t^{(l)} \right) \right\}_{t=1}^{T_l}.$$

During the test phase, one needs to recommend a feasible solution $x_{\text{new}}$ to a new LP problem using the observation of $(A_{\text{new}}, b_{\text{new}}, z_{\text{new}})$ but without the observation of the objective vector $c_{\text{new}}$:

$$(A_{\text{new}}, b_{\text{new}}, z_{\text{new}}) \rightarrow x_{\text{new}}.$$

The focus of the existing literature has been mainly on the risk-neutral objective

$$\min_x \mathbb{E}[c|z]^\top x, \tag{2}$$
$$\text{s.t. } Ax = b, \ x \geq 0$$

where the recommended decision $x$ can be viewed as a function of $(A, b, z)$.

Alternatively, we consider a risk-sensitive objective

$$\min_x \text{VaR}_\alpha[c^\top x | z], \tag{3}$$
$$\text{s.t. } Ax = b, \ x \geq 0$$

where $\alpha \in (0, 1)$. Here $\text{VaR}_\alpha(U)$ denotes the $\alpha$-quantile/value-at-risk (VaR) of a random variable $U$; specifically, $\text{VaR}_\alpha(U) := F_U^{-1}(\alpha)$ with $F_U^{-1}(\cdot)$ be the inverse cumulative distribution function of $U$.

We can interpret the problem (3) in two ways: First, it can be viewed as a risk-sensitive version of the contextual LP problem (2). From the prediction viewpoint, the standard problem (2) concerns only the prediction of $\mathbb{E}[c|z]$, while solving (3) may involve a distributional prediction of the conditional distribution $c|z$. Second, the classic robust LP problem considers a setting where there is no covariate, and the VaR is taken with respect to the (marginal) distribution of $c$. Comparatively, the problem (3) can be viewed as a conditionally robust version where the VaR is taken with respect to the conditional distribution $c|z$. The presence of the covariates $z$ reduces our uncertainty on the objective $c$ and thus will induce less conservative decisions with the same level of risk guarantee.

Throughout the paper, we work with a well-trained ML model $\hat{f} : \mathbb{R}^d \rightarrow \mathbb{R}^n$ that predicts the objective with the covariates and is learned from the training data $\mathcal{D}_l$. We emphasize that we do not impose any assumption on the model, but assume the availability of a validation set $\mathcal{D}_{\text{val}} := \{(c_t, A_t, b_t, z_t)\}_{t=1}^T$ that can be used to quantify the uncertainty of $\hat{f}$.

## 3 Robust Contextual LP

The problem (3) has been extensively studied under the context-free setting in the literature of robust optimization [Ben-Tal et al., 2009], and the key challenge is the non-convexity of the objective function. Equivalently, the problem (3) can be written as the following optimization problem

$$\min_{x, \mathcal{U}} \max_{c \in \mathcal{U}} c^\top x, \tag{4}$$
$$\text{s.t. } Ax = b, \ x \geq 0, \ \mathbb{P}_{c|z}(c \in \mathcal{U}) \geq \alpha$$

where the decision variables are $x$, $c$ and a measurable set $\mathcal{U}$. The last constraint is with respect to the set $\mathcal{U}$, and the probability is taken with respect to the conditional distribution $c|z$. One issue to

solve this problem is the intractability in optimization over the uncertainty set. Particularly, even if the conditional distribution $c|z$ is discrete, (4) is equivalent to a mixed integer linear programming problem, which is generally NP-hard and computationally intensive. Similarly, when solving a *sample average approximation* form of (4) (Bertsimas and Kallus [2020]), the problem becomes similar to the discrete case, and the intractability issue is still inevitable. To resolve this issue, one can solve the following approximation problem with a fixed uncertainty set $\mathcal{U}$ satisfying $\mathbb{P}_{c|z}(c \in \mathcal{U}) \geq \alpha$,

$$\text{LP}(\mathcal{U}) := \min_{x} \max_{c \in \mathcal{U}} c^\top x, \tag{5}$$
$$\text{s.t. } Ax = b, \ x \geq 0.$$

A box- or ellipsoid-shaped uncertainty set enables a tractable optimization of the problem (5). The uncertainty set hopes to cover the high-density region so that the approximation to (3) is tighter.

Now we present two algorithms that implement this approximation idea in the contextual setting. The key is to quantify the uncertainty of the prediction model. Both algorithms construct contextual uncertainty sets for the prediction model; that is, the size of the uncertainty set changes with respect to the covariates $z$. In addition, importantly, the construction of these uncertainty sets accounts for the tractability of the downstream robust optimization problem (5) by outputting the box and ellipsoid shapes, respectively. In both algorithms, we first split the available validation data into two sets $\mathcal{D}_1$ and $\mathcal{D}_2$, and use the first set $\mathcal{D}_1$ for a preliminary calibration, and the second set $\mathcal{D}_2$ for an additional adjustment. The output from both algorithms gives a contextual uncertainty set $\mathcal{U}(z)$ that works as the input of (5) and thus solves the downstream robust optimization problem.

---

**Algorithm 1** Box Uncertainty Quantification (BUQ)

---

1: Input: Dataset $\mathcal{D}_{val}$, ML model $\hat{f}$, parameter $\alpha$
2: Randomly split the validation data into two index sets $\mathcal{D}_1 \cup \mathcal{D}_2 = \{1, ..., T\}$ and $\mathcal{D}_1 \cap \mathcal{D}_2 = \emptyset$
3: *%%Preliminary calibration of the uncertainty sets*
4: **for** $t \in \mathcal{D}_1$ **do**
5:     Calculate the residual vector on the $t$-th validation sample

$$r_t := c_t - \hat{f}(z_t) \tag{6}$$

    and denote $r_t = (r_{t1}, ..., r_{tn})^\top$
6: **end for**
7: Train a quantile regression model $\hat{h}(z) : \mathbb{R}^d \to \mathbb{R}^n$ that minimizes

$$\sum_{t \in \mathcal{D}_1} \sum_{i=1}^{n} \rho_\alpha \left( |r_{ti}| - \hat{h}(z_t)_i \right) \tag{7}$$

    where $\rho_\alpha(\cdot) := \alpha(\cdot)^+ + (1-\alpha)(\cdot)^+$ denotes the pinball loss
8: *%%Additional adjustment of the size of the uncertainty sets*
9: **for** $t \in \mathcal{D}_2$ **do**
10:     Let
$$\bar{c}_t(\eta) := \hat{f}(z_t) + \eta \hat{h}(z_t) \ \in \mathbb{R}^n, \ \ \underline{c}_t(\eta) := \hat{f}(z_t) - \eta \hat{h}(z_t) \ \in \mathbb{R}^n$$
11: **end for**
12: Choose a minimal $\eta > 0$ such that

$$\sum_{t \in \mathcal{D}_2} 1\{\underline{c}_t(\eta) \leq c_t \leq \bar{c}_t(\eta)\} \geq \min\{|\mathcal{D}_2|, \alpha(|\mathcal{D}_2| + 1)\} \tag{8}$$

    where $1\{\cdot\}$ is the indicator function and the inequality is required to hold element-wise
13: Output: $\hat{h}, \eta$

---

In Algorithm 1, we first compute the residual/error vector $r_t \in \mathbb{R}^n$ for samples in dataset $\mathcal{D}_1$. Then for each coordinate $i = 1, ..., n$, we fit a quantile regression model $\hat{h}$ that predicts the quantile of the absolute error by minimizing (7). Specifically, the model $\hat{h}$ aims to predict the quantiles of each coordinate of the residual vector $r_t$. Then, in the second step of Algorithm 1, we pretend the quantile model $\hat{h}$ is "correct" and construct the uncertainty sets accordingly for samples in dataset

$\mathcal{D}_2$. Importantly, the uncertainty set is parameterized by a scalar $\eta$. We use the parameter $\eta$ to perform an additional adjustment of the uncertainty set so that the coverage probability $\alpha$ is met on $\mathcal{D}_2$ by (8). The algorithm outputs the function $\hat{h}$ and $\eta$, and it will output a box-shaped uncertainty set $\mathcal{U}_\alpha^{(1)}(z) = \left[ \hat{f}(z) - \eta\hat{h}(z), \hat{f}(z) + \eta\hat{h}(z) \right]$ for a new sample with covariates $z$.

---

**Algorithm 2** Ellipsoid Uncertainty Quantification (EUQ)

---

1: Input: Dataset $\mathcal{D}_{val}$, ML model $\hat{f}$, parameter $\alpha$
2: Randomly split the validation data into two index sets $\mathcal{D}_1 \cup \mathcal{D}_2 = \{1, ..., T\}$ and $\mathcal{D}_1 \cap \mathcal{D}_2 = \emptyset$
3: *%% Preliminary calibration of the uncertainty sets*
4: **for** $t \in \mathcal{D}_1$ **do**
5:     Calculate the residual vector on the $t$-th validation sample

$$r_t := c_t - \hat{f}(z_t)$$

6: **end for**
7: Use $\mathcal{D}_1$ to fit a model $\hat{g} : \mathbb{R}^d \to \mathbb{R}$ that predicts the $\alpha$-quantile of $\|r_t\|_2$ using the covariates $z_t$
8: Fit a zero-mean normal distribution for the scaled residual vector

$$\{r_t/\hat{g}(z_t) : t \in \mathcal{D}_1\}$$

   and denote the covariance matrix as $\hat{\Sigma}$
9: *%%Additional adjustment of the size of the uncertainty sets*
10: Choose a minimal $\eta > 0$ such that

$$\sum_{t \in \mathcal{D}_2} 1\left\{ \sqrt{(c_t - \hat{f}(z_t))^\top \hat{\Sigma}^{-1}(c_t - \hat{f}(z_t))} \leq \eta\hat{g}(z_t) \right\} \geq \min\{|\mathcal{D}_2|, \alpha(|\mathcal{D}_2| + 1)\} \quad (9)$$

   where $1\{\cdot\}$ is the indicator function and the inequality is required to hold element-wise.
11: Output: $\hat{g}, \hat{\Sigma}, \eta$

---

Algorithm 2 follows a similar spirit as Algorithm 1. It first computes the residual vector $r_t$ for samples in dataset $\mathcal{D}_1$, but differently from Algorithm 1, it fits a regression model for the norm $\|r_t\|_2$. Then it fits a zero-mean normal distribution for the scaled residual vector. The motivation is that the residual vector (up to a scale $\hat{g}(z)$) follows a multivariate normal distribution. We remark that although the algorithm design is motivated by such a setting of normal distribution, its theoretical guarantee does not rely on this normal structure. The key is the second step of additional calibration: it pretends the uncertainty set is "correct" and uses an additional scalar $\eta$ to make further adjustments such that the coverage probability $\alpha$ is met on $\mathcal{D}_2$ by (9). For a new sample with covariates $z$, the algorithm outputs an ellipsoid-shaped uncertainty set $\mathcal{U}_\alpha^{(2)}(z) = \left\{ c : \sqrt{(c - \hat{f}(z))^\top \hat{\Sigma}^{-1}(c - \hat{f}(z))} \leq \eta\hat{g}(z) \right\}$.

In comparison, for the first step of preliminary calibration, Algorithm 1 quantifies the uncertainty of each coordinate of the residual vector individually, while Algorithm 2 presumes a Gaussian structure. We remark that the two algorithms provide two examples for the design of the preliminary calibration step, but these are not the only two options. As we will see shortly, the second step of additional adjustment is in charge of the coverage guarantee, so the first step can be further modified based on other prior knowledge of the underlying data/optimization problem.

## 3.1 Algorithm Analysis

The following proposition provides a coverage guarantee for both Algorithm 1 and 2 in that the output uncertainty set will cover a new sample with a probability of approximately $\alpha$.

**Proposition 1.** *For a new sample $(c, A, b, z)$ from distribution $\mathcal{P}$, denote the uncertainty sets output from Algorithm 1 and Algorithm 2 by $\mathcal{U}_\alpha^{(1)}(z)$ and $\mathcal{U}_\alpha^{(2)}(z)$, respectively. The following inequalities hold for $k = 1, 2$,*

$$\alpha \leq \mathbb{P}\left( c \in \mathcal{U}_\alpha^{(k)}(z) \right) \leq \alpha + \frac{1}{|\mathcal{D}_2| + 1} \quad (10)$$

*where the probability is with respect to the new sample $(c, A, b, z)$ and dataset $\mathcal{D}_2$.*

The proposition gives the following insights into the two algorithms and the general problem of robust contextual LP. First, the second step (in both algorithms) of the adjustment via the scalar $\eta$ is the key to the coverage guarantee (10), with several advantages. First, as the concentration argument is made with respect to only this second step, the bound is rather tight and free of unnecessary conservation in the uncertainty set construction. Second, it allows full flexibility in the choice of the prediction model $\hat{f}$ and the preliminary calibration model $\hat{h}$ (Algorithm 1) and $\hat{g}$ (Algorithm 2). This reinforces the notion of predict-then-calibrate. The design of the prediction model $\hat{f}$ can be solely targeted on the accuracy, and the coverage guarantee can be deferred and taken care of by the calibration steps such as these two algorithms.

Furthermore, we note the probability in (10) is taken with respect to only the validation data and the new sample. It means Proposition 1 holds even if the training data $D_l$ (for obtaining $\hat{f}$) is not sampled from the distribution $\mathcal{P}$. In other words, the result only requires the validation data $\mathcal{D}_{\text{val}} \sim \mathcal{P}$ and it is robust to a distribution shift of the training distribution when $\mathcal{D}_l \not\sim \mathcal{P}$. In a broader context, when the training data, validation data, and test data come from different distributions, this situation is commonly referred to as the out-of-domain (OOD) problem. We refer to the survey papers Shen et al. [2021], Yang et al. [2021] for a more detailed discussion of this problem. Developing more coverage guarantees with OOD data will require future investigations, and we will list this problem as a future direction.

In addition, the coverage guarantee in Proposition 1 can be extended to a robustness guarantee for the contextual LP problem as in the following corollary.

**Corollary 1.** *For a new sample $(c, A, b, z)$ from distribution $\mathcal{P}$, denote the uncertainty set output from Algorithm 1 or Algorithm 2 by $\mathcal{U}_\alpha(z)$. Let $x^*(z)$ and $OPT(z)$ be the optimal solution and the optimal value of $LP(\mathcal{U}_\alpha(z))$ (5). Then we have*

$$\mathbb{P}\left(c^\top x^*(z) \leq OPT(z)\right) \geq \alpha$$

*where the probability is taken with respect to the new sample $(c, A, b, z)$ and $\mathcal{D}_2$.*

To end this section, we remark that although the coverage guarantee (10) is in a population sense where the probability is taken with respect to both $c$ and $z$, our framework of predict-then-calibrate also has the flexibility to achieve an individual coverage by changing the calibration algorithm under some assumptions. Here, the individual coverage means that the conditional probability $\mathbb{P}\left(c \in \mathcal{U}_\alpha(z)|z\right) = \alpha$ holds for any covariate $z$ as the constraint in (4), where $\mathcal{U}_\alpha(z)$ denotes any given contextualized uncertainty set. However, the above-mentioned assumption to achieve individual coverage cannot be verified from the data prior, and achieving this coverage is, in general, a hard problem, although there have been many attempts in the literature on conformal prediction and model calibration. In addition, we want to emphasize that, compared to achieving individual coverage, the most significant focus of this paper is to introduce the idea of uncertainty quantification to solve robust optimization, which disentangles the prediction from the calibration. As a result, we delay the discussion of the individual coverage to Appendix B.1.

## 3.2 Value of Better Prediction and Contextual Uncertainty Set

Two important components in the framework of predict-then-calibrate are (i) the flexibility in choosing the prediction model $\hat{f}$ and (ii) the contextual uncertainty set $\mathcal{U}_\alpha(z)$. In the following, we illustrate the importance of these two aspects via simple examples.

**The value of better prediction.** Consider a single-variable LP

$$\min_x \; cx \;\; \text{s.t.} \; -1 \leq x \leq 1 \tag{11}$$

where the objective is determined by $c = \sum_{i=1}^d z_i - d \cdot \epsilon$. Here $z_i$ and $\epsilon$ are sampled from an exponential distribution $\text{Exp}(1)$ for all $i = 1, .., d$. We note that $z_i$ can be interpreted as either the covariate itself or some useful latent factor extracted by the prediction model. Consider a prediction model that can only utilize/extract the first $k$ covariates/latent factors $z_{1:k} = (z_1, ..., z_k)$. The optimal prediction model would be $f_k(z_{1:k}) = \sum_{i=1}^k z_i$, and also we assume it has perfect uncertainty quantification denoted by $\mathcal{U}_\alpha(z_{1:k})$. We denote the accordingly robust solution by $x^*_\alpha(z_{1:k})$, and thus all these solutions for $k = 1, ..., d$ meet the coverage guarantee.

**Proposition 2.** *For any $\alpha \in (0.5, 1)$ and $k \geq 1$*

$$\mathbb{P}\left(x_\alpha^*(z_{1:k}) = 0\right) =$$

$$\frac{1}{\Gamma(k)}\left(\gamma\left(k, \max\left\{0, -d\log\left((1-\alpha)\left(1+\frac{1}{d}\right)^{d-k}\right)\right\}\right) - \gamma\left(k, \max\left\{0, -d\log\left(\alpha\left(1+\frac{1}{d}\right)^{d-k}\right)\right\}\right)\right),$$

*and it decreases monotonously with respect to $k$. Here, $\gamma(\cdot, \cdot)$ denotes the lower incomplete gamma function.*

Proposition 2 gives an analytical expression for the probability of the optimal robust solution equal to zero. A zero solution can be viewed as a conservative solution when the prediction model is uncertain about the objective $c$. While we assume the prediction model is equipped with perfect uncertainty quantification, the proposition says that more informative covariates and/or more powerful prediction models will lead to less conservative solutions for the downstream robustness task. The reason is that a better prediction model can make the uncertainty quantification task easier by smoothing the conditional distribution of the residual $r|z$ with respect to $z$. For example, when $k = d$ in Proposition 2, the distributions of residuals are identical and thus very smooth for different covariates, which is easier to quantify than the case with $k = 0$. This reason can also be extended to general cases.

**The value of better calibration.** Consider the LP (11) again but with $c$ specified by

$$c = (\text{sign}(z) + \epsilon)\sqrt{|z|},$$

where $z \in \mathbb{R}$ is sampled uniformly from $[-1, 1]$, and $\epsilon$ is sampled from the uniform distribution on $[-0.5, 0.5]$. In this case, the sign of the objective $c$ can be deterministically determined by the sign of $z$. With perfect knowledge of the underlying uncertainty, the following solution is the optimal robust solution for all $\alpha \in (0, 1)$

$$x^*(z) = -\text{sign}(z).$$

The following proposition gives another uncertainty set with the optimal prediction model and a coverage guarantee but a suboptimal robust solution.

**Proposition 3.** *For $\alpha \geq 1/2$, let the uncertainty set*

$$\mathcal{U}_\alpha(z) = \begin{cases} \left[\sqrt{z} - \frac{1-\sqrt{2-2\alpha}}{2}, \infty\right), & \text{if } z \in [0, 1], \\ \left(-\infty, -\sqrt{-z} + \frac{1-\sqrt{2-2\alpha}}{2}\right], & \text{if } z \in [-1, 0). \end{cases}$$

*The uncertainty set has a coverage guarantee in that $\mathbb{P}(c \in \mathcal{U}_\alpha(z)) = \alpha$. If we solve the optimization problem (5) with the uncertainty set $\mathcal{U}_\alpha(z)$, the following robust solution is obtained*

$$x(z) = \begin{cases} -1, & \text{if } z \geq \frac{3-2\alpha-2\sqrt{2-2\alpha}}{4}, \\ 0, & \text{if } \frac{-3+2\alpha+2\sqrt{2-2\alpha}}{4} \leq z \leq \frac{3-2\alpha-2\sqrt{2-2\alpha}}{4}, \\ 1, & \text{if } z \leq \frac{-3+2\alpha+2\sqrt{2-2\alpha}}{4}. \end{cases}$$

The robust solution induced by the uncertainty set $\mathcal{U}_\alpha(z)$ is unnecessarily conservative compared to the optimal robust solution $x^*(z)$. This highlights the importance of good uncertainty quantification, and in particular, contextualized uncertainty quantification. Specifically, the suboptimality of $x(z)$ arises from that the uncertainty set $\mathcal{U}_\alpha(z)$ in the proposition is not much contextualized with respect to $z$, and this justifies the contextualized uncertainty models $\hat{h}$ and $\hat{g}$ for the preliminary calibration step in Algorithm 1 and Algorithm 2.

In addition to the contextualized uncertainty quantification, we remark the flexibility in choosing the calibration model or uncertainty set can bring us two benefits. First, it has the potential to further reduce decision conservatism by collaborating with new designs of small uncertainty sets. Specifically, we can design predict-then-calibrate algorithms with even small uncertainty sets based on many previous works that focus on designing minimum size uncertainty sets to achieve a given coverage guarantee in the context-free robust optimization or distributional robust optimization literature (Bertsimas et al. [2018], Gupta [2019]). Second, this framework is also applicable to more general optimization problems such as convex programs or multi-stage stochastic programs. In

particular, for these general problems, if the robust VaR formulation has a tractable construction of the uncertainty set in the context-free setting, we can then design a predict-then-calibrate algorithm by constructing the corresponding contextual uncertainty set accordingly. To close this section, we emphasize that our predict-then-calibrate framework behaves more like a plug-in module that can fit into the existing robust optimization literature, and thus, it is potentially capable of handling more problems than the contextual LP loss discussed in this paper.

## 4 Extension to Distributionally Robust Contextual LP

In the previous sections, we study the problem of robust contextual LP and the framework of predict-then-calibrate. Now we turn to a related but different problem – distributionally robust contextual LP – through the lens of predict-then-calibrate. We relate the objectives of robustness and distributional robustness with the two types of uncertainty in the uncertainty quantification literature. We show the paradigm of predict-then-calibrate can also be extended to the distributionally robust contextual LP and obtain a new generalization bound for the problem.

The distributionally robust contextual LP concerns the following problem:

$$\min_x \max_{\mathcal{P}' \in \Xi_\gamma} \mathbb{E}_{\mathcal{P}'}\left[c^\top x\right] = \mathbb{E}_{\mathcal{P}'}\left[c\right]^\top x$$

$$\text{s.t. } Ax \le b, \ x \ge 0$$

where $\Xi_\gamma$ denotes a set of distributions that describes the uncertainty on $c$. Ideally, one hopes that the uncertainty set covers the true conditional distribution $c|z$. Compared to the robust version that concerns the VaR/Quantile of the objective $c$, the formulation here still concerns the risk-neutral objective but accounts for the uncertainty in estimating $\mathbb{E}[c|z]$. The distributionally robust optimization has two roles: first, it can be used to derive solutions with provable guarantees on generalization, and second, it induces a regularization effect for the underlying prediction model.

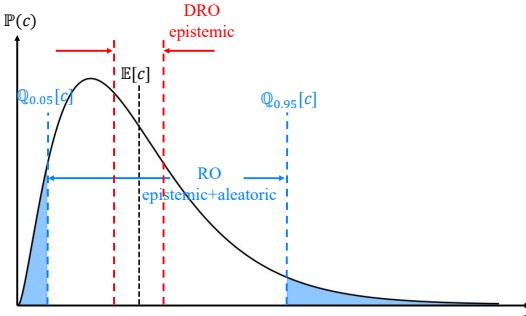

Figure 1: Illustration of robust optimization (RO) and distributionally robust optimization (DRO) (here we omit the covariates $z$ for simplicity). DRO concerns epistemic uncertainty Hüllermeier and Waegeman [2021], the part of the uncertainty that can be reduced by obtaining more information. With more samples, the uncertainty set will shrink. In contrast, aleatoric uncertainty Hüllermeier and Waegeman [2021] refers to the inherent and irreducible randomness of $c$. To derive the uncertainty set for RO problem, one needs to account for the summation of epistemic and aleatoric uncertainty.

As in the previous section, we denote the prediction error on the validation data $\mathcal{D}_{\text{val}}$ by

$$r_t = c_t - \hat{f}(z_t).$$

The following proposition states that there exists a distributionally robust algorithm that achieves a generalization bound that is oblivious with respect to the underlying prediction model $\hat{f}$.

**Proposition 4.** *Assume there exists a constant $L$ such that the following condition holds for any $z, z'$*

$$|\mathbb{E}[r|z] - \mathbb{E}[r|z']| \le L\|z - z'\|_2^s.$$

*Then under mild boundedness assumptions on the distribution, there exists a distributionally robust optimization algorithm that utilizes the validation data $\mathcal{D}_{val}$ and outputs a solution $\hat{x}(z)$ such that the following inequality holds with high probability*

$$\mathbb{E}\left[c^\top\left(\hat{x}(z) - x^*(z)\right)\right] \le O\left(T^{-\frac{s}{2(s+d)}}\log T\right).$$

We defer the detailed proof and the DRO algorithm to Appendix C.5. Specifically, the DRO algorithm performs a nonparametric uncertainty quantification with respect to $r_t$'s and thus it is oblivious to the underlying prediction model $\hat{f}$. We remark that such idea of calibrating $r_t$'s is not new and has been exploited in Wang et al. [2021], Kannan et al. [2020, 2021]. But notably, our result requires a weaker condition on the underlying $\hat{f}$, where the previous works rely on a realizability condition of the prediction model. Moreover, compared with the generalization bound in the contextual LP literature Liu and Grigas [2021], although Liu and Grigas [2021] develops an $O(T^{-1/4})$ generalization bound, they still require the function class of the prediction model can have bounded Rademacher complexity. If that condition holds here, our generalization bound can also be improved accordingly.

## 5 Experiment

In this section, we illustrate the performance of our proposed algorithms via one simple example and also a shortest path problem considered in Elmachtoub and Grigas [2022] and Hu et al. [2022]. We defer more results including more visualizations, another experiment on the fractional knapsack problem Ho-Nguyen and Kılınç-Karzan [2022], and experiment setups to the Appendix. We implement our Algorithm 1 (PTC-B) and Algorithm 2 (PTC-E) against several benchmark methods including the Ellipsoid method that ignores the covariates, $k$-nearest neighbor (kNN) approach in Ohmori [2021], the Deep Cluster-Classify (DCC), and Integrated Deep Cluster-Classify (IDCC) approaches in Chenreddy et al. [2022]. More experiment details are deferred to Appendix A.

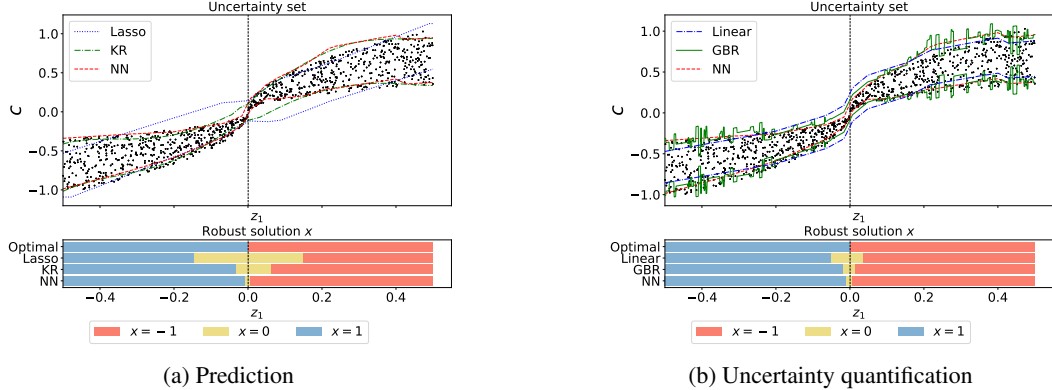

(a) Prediction          (b) Uncertainty quantification

Figure 2: Performance under different prediction and quantile regression models in PTC-B. The results corresponding to PTC-E are the same since PTC-B and PTC-E coincide in this one-dimensional case. The scattered black points indicate the training samples and the curves indicate the band for the uncertainty set. For the left panel, we fix the uncertainty quantification model $\hat{h}$ to be a neural network and implement different prediction models. For the right panel, we fix the prediction model $\hat{f}$ as a neural network and implement different uncertainty quantification models. KR stands for kernel regression, GBR stands for gradient boosting regression, and NN stands for neural network.

**Simple LP Visualization**

In the first experiment, we study the LP problem (11) where the optimal solution and the uncertainty set can be visualized. Here the covariates $z = (z_1, ..., z_d)^\top$ where $z_i$ is sampled from Unif$[-0.5, 0.5]$ independently for $i = 1, ..., d$, $\epsilon \sim$ Unif$[-0.5, 0.5]$, and $c = (\text{sign}(z_1) + \epsilon) \cdot \sqrt{|z_1|}$ (c is independent of $z_2, ..., z_d$). We consider a risk level of $\alpha = 0.8$ and dataset size $T = 1000$. For our methods of PTC-B and PTC-E, we use 60% of the data for training $\hat{f}$, 20% for preliminary calibration ($\mathcal{D}_1$), and 20% for final adjustment ($\mathcal{D}_2$). In Figure 2, the robust solutions obtained from different prediction methods and different calibration methods are plotted. The results reinforce the discussions in Section 3.2 that either a better prediction model or a better calibration model will improve the final performance of the robust optimization problem.

**Shortest path problem**

Now we present numerical experiments for a shortest path problem on a $5 \times 5$ grid with 40 edges, and the cost of traveling through edge $i$ is $c_i$ for $i = 1, ..., n$. We follow the experiment setup in

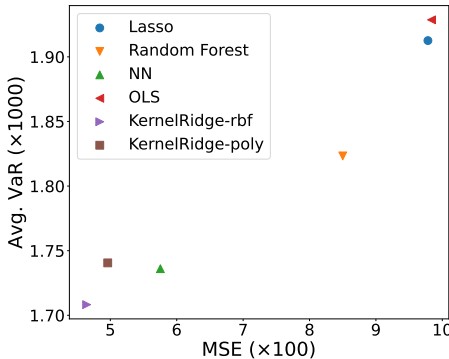
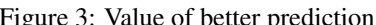

Figure 3: Value of better prediction

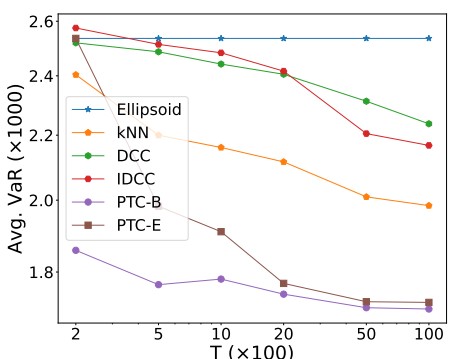

Figure 4: Sample efficiency

Table 1: Average VaR and coverage on shortest path problems under different risk level settings. The average is reported based on 500 simulation trials.

<table>
<tr><td colspan="7">(a) Average VaR</td></tr>
<tr><td>$\alpha$</td><td>Ellips.</td><td>kNN</td><td>DCC</td><td>IDCC</td><td>PTC-B</td><td>PTC-E</td></tr>
<tr><td>0.6</td><td>2447</td><td>1940</td><td>2218</td><td>2181</td><td>1650</td><td>1690</td></tr>
<tr><td>0.7</td><td>2488</td><td>1973</td><td>2273</td><td>2168</td><td>1683</td><td>1717</td></tr>
<tr><td>0.8</td><td>2535</td><td>2010</td><td>2312</td><td>2205</td><td>1708</td><td>1774</td></tr>
<tr><td>0.85</td><td>2563</td><td>2032</td><td>2286</td><td>2286</td><td>1735</td><td>1796</td></tr>
<tr><td>0.9</td><td>2598</td><td>2059</td><td>2392</td><td>2291</td><td>1769</td><td>1843</td></tr>
<tr><td>0.95</td><td>2646</td><td>2098</td><td>2373</td><td>2249</td><td>1832</td><td>1905</td></tr>
</table>

<table>
<tr><td colspan="7">(b) Average coverage</td></tr>
<tr><td>$\alpha$</td><td>Ellips.</td><td>kNN</td><td>DCC</td><td>IDCC</td><td>PTC-B</td><td>PTC-E</td></tr>
<tr><td>0.6</td><td>0.63</td><td>0.65</td><td>0.62</td><td>0.63</td><td>0.56</td><td>0.63</td></tr>
<tr><td>0.7</td><td>0.73</td><td>0.65</td><td>0.7</td><td>0.73</td><td>0.64</td><td>0.71</td></tr>
<tr><td>0.8</td><td>0.82</td><td>0.65</td><td>0.8</td><td>0.84</td><td>0.76</td><td>0.8</td></tr>
<tr><td>0.85</td><td>0.87</td><td>0.65</td><td>0.85</td><td>0.89</td><td>0.79</td><td>0.85</td></tr>
<tr><td>0.9</td><td>0.92</td><td>0.65</td><td>0.89</td><td>0.93</td><td>0.88</td><td>0.91</td></tr>
<tr><td>0.95</td><td>0.96</td><td>0.65</td><td>0.94</td><td>0.97</td><td>0.93</td><td>0.97</td></tr>
</table>

Elmachtoub and Grigas [2022] and Hu et al. [2022] where the details are deferred to Appendix A.2. In this experiment, we demonstrate several aspects of our algorithms. First, Figure 3 shows the averaged VaR of PTC-B solutions (under different prediction models $\hat{f}$) versus the Mean Squared Error (MSE) on the test data, which reveals a positive correlation between the predictive performance and the quality of the solution obtained in the downstream robust optimization task. We utilize the Kernel Ridge method with the RBF kernel—identified as the top-performing model in Figure 3—as the predictive model and the Neural Network (NN) model as the preliminary calibration model for both PTC-B and PTC-E in the ensuing experiments (Figure 4 and Table 1) to compare with other algorithms. Second, Figure 4 demonstrates that our algorithms exhibit better sample efficiency than the benchmark algorithms as we increase the number of training samples. Specifically, while the model DCC and IDCC take advantage of the expressiveness of a complicated deep learning architecture, they can be more costly in terms of the required training samples. Furthermore, we compare the performance under varying risk level settings in Table 1. The Ellipsoid method does not utilize the covariates information, so although it has a coverage guarantee, it performs worst on the VaR. The kNN method [Ohmori, 2021] has a good VaR performance, but it lacks confidence adjustment, so it is hard to adapt to different confidence levels. The two deep learning-based approaches and ours give good coverage guarantees, while we attribute the advantage of our algorithms to the flexibility in choosing the prediction model and the calibration model.

**Conclusion.** In this paper, we study the robust contextual LP problem which lies at the intersection of contextual LP/predict-then-optimize and robust optimization. We develop two algorithms for the problem which output box- and ellipsoid shape uncertainty sets. The algorithms introduce a new perspective for contextualized uncertainty calibration, and it highlights the convenience brought by the disentanglement of the prediction and the uncertainty calibration. Yet, as mentioned earlier, the coverage guarantee in Proposition 1 is with respect to a population/average sense. Developing algorithms with individual-level coverage guarantees deserves more future investigation. Also, we hope our work provides a starting point for future research on risk-sensitive and robust decision-making for the problem of contextual optimization and optimization with machine-learned advice.

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

# A Supplementary for Experiment

**Details of benchmarks.** For the DCC and IDCC algorithms proposed in Chenreddy et al. [2022], we set the number of clusters to be 10. This choice of the number of clusters is large enough because there are clusters containing no training samples. The Ellipsoid method, which is also a benchmark used in Chenreddy et al. [2022], fits a Gaussian distribution without contextual information and calibrates the radius to cover a proportion of $\alpha$ training samples. For the $k$NN conditional RO method [Ohmori, 2021], the hyperparameter, $k$, lacks the flexibility to adjust with respect to the number of training samples $T$, the dimensionality of the problem, and the risk level $\alpha$, so we empirically choose $k = \max\{\sqrt{T}, 2 \times n\}$ here. Different algorithms utilize the same dataset to ensure fairness in each round of comparison. We redefine $T$ as its size in the experiment section. For our proposed Algorithm 1 (PTC-B) and 2 (PTC-E) under the PTC framework, the dataset is randomly split into 60%, 20%, and 20% to do prediction, preliminary calibration, and final adjustment, respectively.

**Performance measurements.** In the test phase, we randomly generate $z$ from its marginal distribution. Then, for each $z$, we generate a set with 1000 objective vectors, $\{c^t\}_{t=1}^{1000}$, from $\mathbb{P}_{c|z}$. Given any robust solution $x$ conditioned on $z$, we estimate the $\text{VaR}_\alpha[c^\top x|z]$ by the $\alpha$-quantile of $\{c^{t\top}x\}_{t=1}^{1000}$ and take its empirical expectation over $z$ as the average VaR. Similarly, we estimate the average coverage by the average frequency of $\{c \in \mathcal{U}_z\}$.

## A.1 Simple LP Visualization

We select NN as the prediction and preliminary calibration models in both PTC-B and PTC-E algorithms and compare them to benchmarks. Here, because $c$ is only 1-dimensional, we replaced the DCC and IDCC algorithms, which are based on deep neural networks, with another clustering method, K-Means. Figure 5 shows the performance of different algorithms when $d$ increases ($d = 1$, 4, and 16), and the plot is shown by fixing $z_2, ..., z_d$ to 0 in the test phase. The Ellipsoid method performs the worst and gives an unchanged uncertainty set for all different $z$ due to its ignorance of the contextual information. All algorithms degrade with an increase in irrelevant information, but localized methods such as kNN and K-Means exhibit more severe deterioration. Our proposed methods, however, maintain a certain degree of stability.

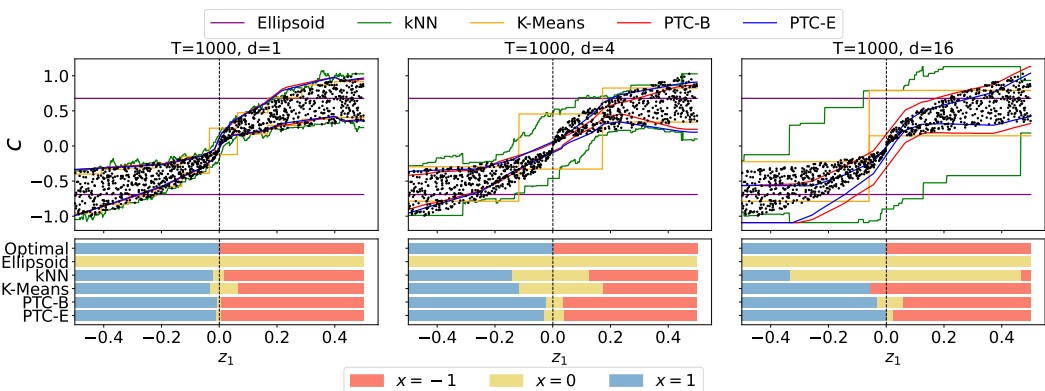

Figure 5: Comparison in performance deterioration with increasing dimensions

## A.2 Shortest Path Problem

In this section, we consider a shortest path problem on a $5 \times 5$ grid with 40 edges, and the cost of traveling through edge $i$ is $c_i$ for $i = 1, ..., 40$. After observing the covariates $z$ (e.g., weather conditions, day of the week), we aim to select a route from the left-top vertex to the right-bottom one to minimize the value-at-risk travel time, i.e., $\text{VaR}_{\alpha,z}(c^\top x)$. The data generation process is as follows. First, we generate a 0-1 matrix $\Theta \in \mathbb{R}^{40 \times d}$ once with random seed 0 and fix it to encode the parameters of the true model, where each entry is generated from a Bernoulli distribution with probability 0.5. Then, the covariate vector $z^t \in \mathbb{R}^d$ is generated from $N(0, I_d)$ for $t = 1, ..., T$.

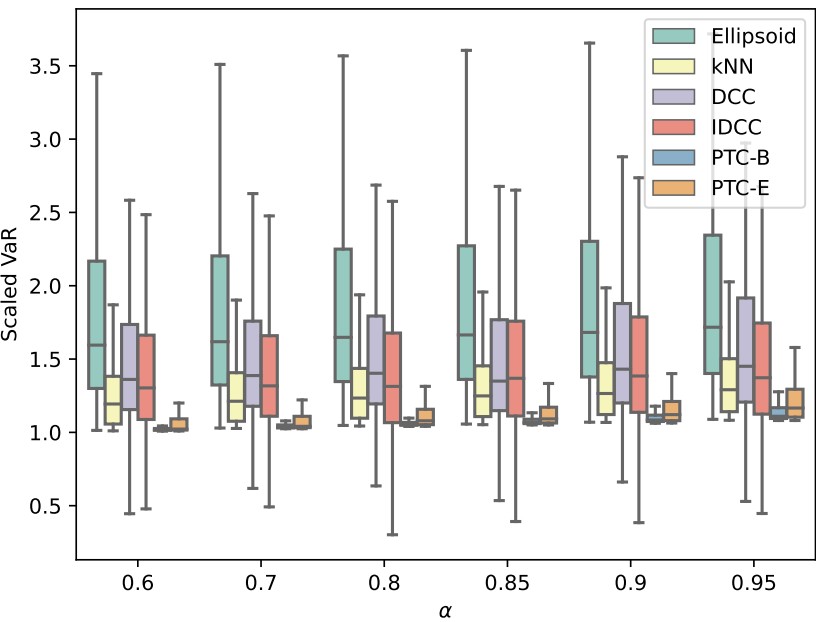

Figure 6: Box plot of scaled VaR on shortest path problems under different risk level settings

Given $\boldsymbol{z}^t$, the cost on edge $i$ is $c_i^t = [(\frac{1}{\sqrt{d}}(\Theta \boldsymbol{z}^t) + 3)^5 + 1] \cdot \epsilon_i$, where $\epsilon_i \sim \mathrm{Unif}[\frac{3}{4}, \frac{5}{4}]$ independently for $i = 1, ..., 40$. We implement $d = 10$ here. To simulate the uneliminated irrelevant information, we set the last two dimensions of $z$ to be independent of $c$, so their corresponding column vectors in $\Theta$ are set to be zero. In the test phase, we evaluate the performance on $500$ $z$'s generated from the marginal distribution of $z$. We select the Kernel Ridge method with the RBF kernel as our prediction model and the NN model as our preliminary calibration model in both PTC-B and PTC-E.

In Figure 6, we show the box plot version of Table 1 with scaled VaR. Here, to alleviate the impact of randomness on $z$, we scale the VaR by the optimal value of the corresponding conditional stochastic program with expectation objective, i.e., $\mathrm{VaR}_\alpha/\mathrm{Obj}^E$, where $\mathrm{Obj}^E = \max_{Ax=b,x\geq 0} \mathbb{E}[c|z]^\top x$. Figure 6 once again demonstrates the superior performance of our algorithms in minimizing VaR as that in Table 1.

### A.3 Fractional Knapsack Problem

In this section, we consider a fractional knapsack problem and follow the setup in Ho-Nguyen and Kılınç-Karzan [2022]. This problem can be interpreted as a maximization problem, wherein a consumer aims to maximize their utility by selecting items under a budget constraint. Specifically, to fit our formulation, the problem can be written as

$$\min_{x \in \mathcal{X}} \mathrm{VaR}_{\alpha,z}(-c^\top x), \text{ where } \mathcal{X} := \{x \in [0,1]^n : p^\top x \leq B\}.$$

Here, $c$ is the utility vector and we use a minimization version to represent a maximization problem. $p \in \mathbb{R}^n$ is some fixed positive vector indicating the price of items, and $B > 0$ is the budget. The same as the shortest path problem, we randomly generate a 0-1 matrix $\Theta \in \mathbb{B}^{20 \times d}$ once with the last 2 columns entirely filled as zeros to indicate the independence of the last dimensions of $z$ to $c$. Then, we generate multivariate independent $z$ from $\mathrm{Unif}[0,4]^d$, and $c = (\Theta z)^2 \cdot \epsilon$, where $\epsilon$ is multivariate independent and following $\mathrm{Unif}[\frac{4}{5}, \frac{6}{5}]^n$. We conduct experiments on data generated under $T = 5000$, $d = 10$, and $n = 20$, and we choose the Kernel Ridge method with RBF kernel as our prediction model, and the NN as our preliminary calibration model in both the PTC-B and PTC-E.

Given a new covariate $z$, the conditional RO algorithm will output an uncertainty set. Under different constraints, the worst-case scenario corresponding to the uncertainty set may vary, so we compare the performance under randomly generated constraints. We generate the price vector $p$ with each entry as an integer uniformly sampled from $[1, 1000]$. Then, we randomly generated $u \sim \mathrm{Unif}[0,1]$, and the

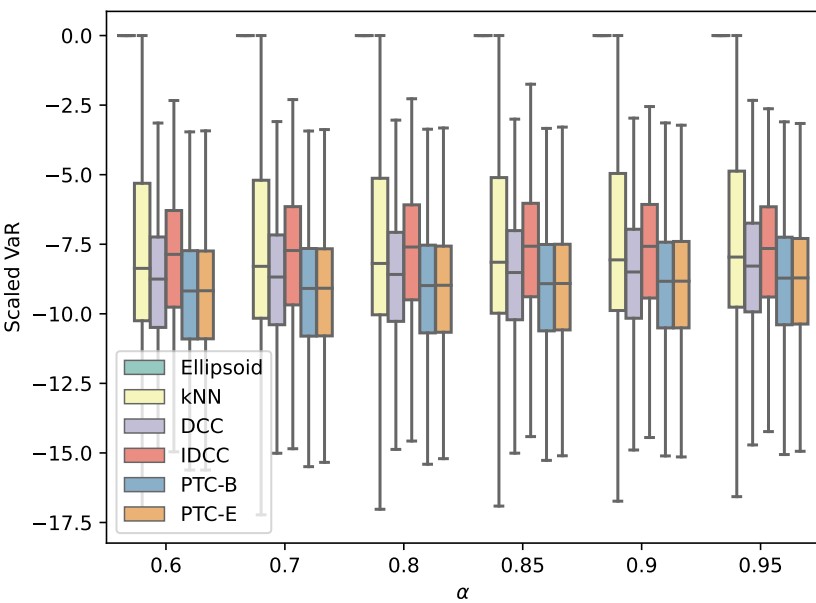

Figure 7: Box plot of scaled VaR on knapsack problems under different risk level settings

budget $B$ is sampled from $\text{Unif}[\max_j p_j, \mathbf{1}^\top p - u \cdot \max_j p_j]$. Here, we randomly generated 10 sets of constraints $\{(p^i, B^i)\}_{i=1}^{10}$ (with random seed 0) and fix them. In the test phase, the average VaR and coverage are taken over 100 $z$'s and 10 constraints (1000 instances in total).

Table 2 associated with Figure 7 displays the comparison results under varying risk levels and demonstrates the advantages of our proposed algorithms again via the lower VaR and proper coverage that closely matches the risk level, where the scaled VaR in Figure 7 is the VaR divided by $\max_{x \in \mathcal{X}} \mathbb{E}[c|z]^\top x$. The contextual-ignorance method, Ellipsoid, performs the worst with the total utilities remaining 0, which shows its excessive conservatism by specifying the worst-case utility of each item to be non-positive and selecting nothing. The DCC and IDCC, as well as our PTC-B and PTC-E methods, exhibit better performance than the kNN method due to our utilization of expressive NN models, which are not limited to utilizing only local information. By virtue of the flexibility of our algorithms in the model selection, the chosen Kernel Ridge regression model is more suitable than the clustering-based DCC and IDCC methods for this problem with continuous covariates, so our algorithms achieve relatively better performance than theirs.

Table 2: Average VaR and coverage on fractional knapsack problems under different risk level settings. The result is reported based on 500 simulation trials.

(a) Average VaR

| $\alpha$ | Ellips. | kNN | DCC | IDCC | PTC-B | PTC-E |
|---|---|---|---|---|---|---|
| 0.6 | 0 | -1175 | -1256 | -1131 | -1310 | -1310 |
| 0.7 | 0 | -1148 | -1241 | -1111 | -1298 | -1297 |
| 0.8 | 0 | -1137 | -1230 | -1090 | -1283 | -1282 |
| 0.85 | 0 | -1128 | -1223 | -1086 | -1273 | -1273 |
| 0.9 | 0 | -1114 | -1213 | -1087 | -1261 | -1261 |
| 0.95 | 0 | -1102 | -1180 | -1092 | -1242 | -1243 |

(b) Average coverage

| $\alpha$ | Ellips. | kNN | DCC | IDCC | PTC-B | PTC-E |
|---|---|---|---|---|---|---|
| 0.6 | 0.66 | 0.96 | 0.75 | 0.65 | 0.57 | 0.65 |
| 0.7 | 0.74 | 0.96 | 0.79 | 0.74 | 0.73 | 0.71 |
| 0.8 | 0.82 | 0.96 | 0.86 | 0.83 | 0.79 | 0.82 |
| 0.85 | 0.86 | 0.96 | 0.89 | 0.89 | 0.88 | 0.86 |
| 0.9 | 0.91 | 0.96 | 0.92 | 0.93 | 0.91 | 0.91 |
| 0.95 | 0.96 | 0.96 | 0.97 | 0.95 | 0.92 | 0.97 |

# B   Supplementary for Section 3

This section provides discussions about the individual coverage and proofs for theoretical statements in Section 3. In Section B.1, we will provide a brief literature review about global coverage and individual coverage, and then, provide an algorithm to achieve the individual coverage with

corresponding theoretical results. In Sections B.2, B.3, and B.4, we will show Propositions 1, 2, 3, respectively. The proof will utilize some useful lemmas in Section D.

## B.1 Individual Coverage

The uncertainty quantification literature arises from two communities. It is referred to as "conformal prediction" in the field of statistics and "model calibration" in the field of machine learning. In the following, we briefly review the results of global and individual coverage in uncertainty quantification in those two communities, respectively.

**Conformal prediction:** Milestone works on the topic, Lei et al. [2018], Barber et al. [2023], focus on global coverage. Angelopoulos and Bates [2021], a survey paper on conformal prediction, also focuses on the global guarantee. For conditional/individual coverage, Foygel Barber et al. [2021], Vovk [2012] state an "impossible triangle" in uncertainty quantification: (i) the coverage result is built for conditional coverage; (ii) it makes no assumptions on the underlying distribution; (iii) it has finite-sample guarantee rather than asymptotic consistency. Essentially, in our paper, Algorithms 1 and 2 basically fail to meet (i), but they barely rely on any assumptions on the underlying distribution and enjoy a finite-sample guarantee, that is, achieving (ii) and (iii). Later in this section, we will introduce a nonparametric algorithm that achieves conditional coverage, and the coverage can be finite-sample, i.e., achieving (i) and (iii), while this result requires additional assumptions on the underlying distribution, which is a violation of (ii).

**Model calibration/recalibration:** Lakshminarayanan et al. [2017], Cui et al. [2020], Chung et al. [2021], Kuleshov et al. [2018], Zhao et al. [2020] are a few state-of-the-art papers on calibrating the uncertainty of a regression model. We note that none of these papers claim to achieve a conditional coverage guarantee. Rather, they ensure a global guarantee and strive for better empirical conditional coverage. In particular, Zhao et al. [2020] gives a pessimistic result on conditional coverage that it is impossible to verify whether an algorithm achieves individual coverage. However, the proof is based on a very special case, and it is possible to rule it out with some mild conditions.

In the following, we present Algorithm 3 under our predict-then-calibrate framework, which can achieve individual coverage.

---

**Algorithm 3** Uncertainty Quantification with Individual Coverage

---

1: Input: Validation data $\mathcal{D}_{val}$, ML model $\hat{f}$, kernel choice $K_h(\cdot, \cdot)$, bandwidth $h$, target covariate $z_0$

2: For each $t \in \mathcal{D}_{val}$, let
$$r_t := c_t - \hat{f}(z_t)$$

3: Define the estimation of the conditional distribution $r|z = z_0$:
$$\hat{P}_{r|z_0} := \sum_{t \in \mathcal{D}_{val}} \frac{K_h(z_t, z_0)\delta_{r_t}}{\sum_{t \in \mathcal{D}_{val}} K_h(z_t, z_0)},$$

where $\delta_r$ denotes the delta distribution on $r$

4: Choose a minimal $\eta > 0$ such that
$$\hat{P}_{r|z_0}\left(\{r : \|r\|_2 \leq \eta\}\right) \geq \alpha$$

5: Output: Uncertainty set $\mathcal{U}_\alpha(z_0) = \left\{c : \|c - \hat{f}(z_0)\|_2 \leq \eta\right\}$

---

Here, this uncertainty quantification algorithm is mainly based on a nonparametric estimation in Hall et al. [1999], Liu et al. [2023], which is also similar to the nonparametric estimation in Algorithm 4. The estimator in Step 3 in Algorithm 3 covers a wide range of non-parametric estimators with different kernels, including $k$-nearest neighbors, and it can approximate the conditional distributions of the residuals $r|z = z_0$ with a sublinear convergence rate under some Lipschitz conditions. As a result, by using this estimator, we can directly calculate the size of the uncertainty set so that the estimated conditional distribution attains individual coverage. That is Step 4 in Algorithm 3. In the

following, we will provide a statement without proof since the analysis is almost the same as the proof of Proposition 4. We also refer to Hall et al. [1999], Liu et al. [2023] for a detailed analysis.

**Proposition 5.** *Assume there exists a constant $L'$ such that the following Lipschitz condition holds for any $z, z'$*

$$TV(P_{r|z}, P_{r|z'}) \leq L\|z - z'\|_2^s,$$

*where $TV(\cdot, \cdot)$ denotes the total variation distance between two distributions, and $P_{r|z}, P_{r|z'}$ denotes the conditional distribution of the residuals given covariates $z$ or $z'$, respectively. Then under mild boundedness assumptions on the distribution, the uncertainty set $\mathcal{U}_\alpha(z_0)$ output from Algorithm 3 satisfies*

$$TV(\hat{P}_{r|z_0}, P_{r|z_0}) \leq O\left(T^{-\frac{s}{2(s+d)}} \log T\right).$$

Moreover, it is worth noting that, through this approximation of the conditional distributions, we can extend our framework in tractable optimizing conditional value-at-risk besides VaR in (3). Specifically, one can generate a number of independent samples $\{\tilde{c}_k\}_{k=1}^K$ from the approximation of the conditional distribution $c|z = z_0$. Then to optimize the following empirical CVaR objective (Rockafellar et al. [2000])

$$\min_{x, \gamma} \gamma + \frac{1}{K(1-\alpha)} \sum_{k=1}^K \left(\tilde{c}_t^\top x - \gamma\right)^+$$

$$\text{s.t. } Ax = b, \ x \geq 0.$$

This is a convex program that has an equivalent linear program and is thus computationally tractable.

To close this section, we want to emphasize that the Lipschitz assumption in Proposition 5 cannot be verified from the data prior. Also, we prefer not to claim we have solved the conditional coverage problem for two reasons: (i) We do not want to give an impression to the robust optimization community that conditional coverage is an easy task to achieve. One should proceed with caution when applying the nonparametric algorithm; (ii) We don not want to give an impression to the statistics and ML communities working on uncertainty quantification problems that we underestimate the difficulty of the conditional coverage and lack of understanding of the related literature.

## B.2 Proof for Proposition 1 and Corollary 1

In this section, we first show Proposition 1, then show Corollary 1 based on the result of Proposition 1.

**Proposition 1.** *For a new sample $(c, A, b, z)$ from distribution $\mathcal{P}$, denote the uncertainty sets output from Algorithm 1 and Algorithm 2 by $\mathcal{U}_\alpha^{(1)}(z)$ and $\mathcal{U}_\alpha^{(2)}(z)$, respectively. The following inequalities hold for $k = 1, 2$*

$$\alpha \leq \mathbb{P}\left(c \in \mathcal{U}_\alpha^{(k)}(z)\right) \leq \alpha + \frac{1}{|\mathcal{D}_2| + 1}$$

*where the probability is with respect to the new sample $(c, A, b, z)$ and dataset $\mathcal{D}_2$.*

*Proof.* Denote $n = |\mathcal{D}_2|$ as the number of samples in the $\mathcal{D}_2$. In the following, we show for $k = 1, 2$,

$$\mathbb{P}\left(c \in \mathcal{U}_\alpha^{(k)}(z)\right) = \frac{\lceil \alpha(n+1) \rceil}{n+1}, \tag{12}$$

where $\lceil \cdot \rceil$ denotes the ceiling function. If it holds, we can prove this proposition by the following inequalities

$$\frac{\lceil \alpha(n+1) \rceil}{n+1} \geq \frac{\alpha(n+1)}{n+1} = \alpha,$$

$$\frac{\lceil \alpha(n+1) \rceil}{n+1} \leq \frac{\alpha n + 2}{n+1} = \alpha + \frac{1}{n+1}.$$

To show (12), we will use the exchangeability of the dataset $\mathcal{D}_2$ and new sample $(c, A, b, z)$. Specifically, denote $\{c_t\}_{t=1}^n$ as the samples in $\mathcal{D}_2$, and denote $c_{n+1}$ as the new sample. Then, we define $\eta_t$

for $t = 1, ..., n + 1$ with respect to Algorithms 1 or 2 as follows

$$\eta_t = \begin{cases} \min \left\{ \eta \geq 0 : \underline{c}_t(\eta) \leq c_t \leq \bar{c}_t(\eta) \right\}, & \text{if } k = 1, \\ \min \left\{ \eta \geq 0 \sqrt{(c_t - \hat{f}(z_t))^\top \hat{\Sigma}^{-1} (c_t - \hat{f}(z_t))} \leq \eta \hat{g}(z_t) \right\}, & \text{if } k = 2. \end{cases}$$

In the proof below, we only show (12) for Algorithm 1, and the proof for Algorithm 2 can be obtained similarly. Based on the choice of the uncertainty set and definition of $\eta$ in Algorithm 1, we have that $\eta$ is the $\frac{\lceil \alpha(n+1) \rceil}{n}$-upper quantile in $\{\eta_t\}_{t=1}^n$. This choice of $\eta$ also implies $c_{t+1} \in \mathcal{U}_\alpha^{(1)}$ if $\eta_{n+1} \leq \eta$, which means

$$\mathbb{P}\left( c_{n+1} \in \mathcal{U}_\alpha^{(1)} \right) = \mathbb{P}\left( \eta_{n+1} \text{ is among the } \lceil \alpha(n+1) \rceil\text{-smallest of } \{\eta_t\}_{t=1}^{n+1} \right). \tag{13}$$

Then, it is sufficient to show

$$\mathbb{P}\left( \eta_{n+1} \text{ is among the } \lceil \alpha(n+1) \rceil\text{-smallest of } \{\eta_t\}_{t=1}^{n+1} \right) = \frac{\lceil \alpha(n+1) \rceil}{n+1}.$$

To this end, because $\{c_t\}_{t=1}^{n+1}$ are i.i.d., we have

$$\mathbb{P}\left( \eta_{n+1} \text{ is among the } \lceil \alpha(n+1) \rceil\text{-smallest of } \{\eta_t\}_{t=1}^{n+1} \right) \tag{14}$$
$$= \mathbb{P}\left( \eta_t \text{ is among the } \lceil \alpha(n+1) \rceil\text{-smallest of } \{\eta_t\}_{t=1}^{n+1} \right)$$

for all $t = 1, ..., n,$, which is also referred to as the exchangability. Then, we can finish the proof by the following equalities:

$$\mathbb{P}\left( \eta_{n+1} \text{ is among the } \lceil \alpha(n+1) \rceil\text{-smallest of } \{\eta_t\}_{t=1}^{n+1} \right)$$
$$= \frac{1}{n+1} \sum_{t=1}^{n+1} \mathbb{P}\left( \eta_t \text{ is among the } \lceil \alpha(n+1) \rceil\text{-smallest of } \{\eta_t\}_{t=1}^{n+1} \right)$$
$$= \frac{1}{n+1} \mathbb{E}\left[ \sum_{t=1}^{n+1} \mathbf{1} \left\{ \eta_t \text{ is among the } \lceil \alpha(n+1) \rceil\text{-smallest of } \{\eta_t\}_{t=1}^{n+1} \right\} \right]$$
$$= \frac{\lceil \alpha(n+1) \rceil}{n+1},$$

where the first equality is obtained by the exchangeability (14), the second equality is obtained by the definition of the expectation, and the last equality is obtained by the fact that

$$\sum_{t=1}^{n+1} \mathbf{1} \left\{ \eta_t \text{ is among the } \lceil \alpha(n+1) \rceil\text{-smallest of } \{\eta_t\}_{t=1}^{n+1} \right\} = \lceil \alpha(n+1) \rceil.$$

$\square$

Next, we show Corollary 1:

**Corollary 1.** *For a new sample $(c, A, b, z)$ from distribution $\mathcal{P}$, denote the uncertainty set output from Algorithm 1 or Algorithm 2 by $\mathcal{U}_\alpha(z)$. Let $x^*(z)$ and $OPT(z)$ be the optimal solution and the optimal value of $LP(\mathcal{U}_\alpha(z))$ (5). Then we have*

$$\mathbb{P}\left( c^\top x^*(z) \leq OPT(z) \right) \geq \alpha$$

*where the probability is with respect to the new sample $(c, A, b, z)$ and $\mathcal{D}_2$.*

*Proof.* By definition of the robust optimization problem (5), we have

$$c^\top x^*(z) \leq OPT(z)$$

if $c \in \mathcal{U}_\alpha^{(k)}(z)$ for $k = 1, 2$. Then, Proposition 1 implies

$$\mathbb{P}\left( c^\top x^*(z) \leq OPT(z) \right) \geq \alpha.$$

$\square$

### B.3 Proof for Proposition 2

**Proposition 2.** *For any $\alpha \in (0.5, 1)$*

$$\mathbb{P}\left(x_\alpha^*(z_{1:k}) = 0\right) =$$

$$\frac{1}{\Gamma(k)} \left( \gamma \left( k, \max\left\{0, -d\log\left((1-\alpha)\left(1+\frac{1}{d}\right)^{d-k}\right)\right\} \right) - \gamma \left( k, \max\left\{0, -d\log\left(\alpha\left(1+\frac{1}{d}\right)^{d-k}\right)\right\} \right) \right),$$

*and it decreases monotonously with respect to $k$.*

*Proof.* Recall that

$$c = \sum_{i=1}^{d} z_i - d\epsilon, \tag{15}$$

and $z_i$ and $\epsilon$ are drawn from an Exponential distribution $\text{Exp}(1)$ for all $i = 1, ..., d$. For any fixed $k = 1, ..., d$, the corresponding prediction model is $f_k(z_{1:k}) = \sum_{i=1}^{k} z_i$.

We first characterize the distribution of the objective vector $c$ given $z_{1:k}$ for any $k = 1, ..., d$. By the additivity of Gamma distributions and the fact that the exponential distribution is a special case of the Gamma distribution, we have given $z_{1:k}$

$$
\begin{aligned}
\mathbb{P}(c \geq 0) &= \int_0^\infty ... \int_0^\infty 1\left\{f_k(z_{1:k}) + \sum_{i=k+1}^{d} z_i \geq d \cdot \epsilon\right\} \cdot \prod_{i=k+1}^{d} e^{-z_i} \cdot e^{-\epsilon} d\epsilon dz_{k+1}...dz_d \\
&= \int_0^\infty \int_0^\infty 1\{f_k(z_{1:k}) + \tilde{z} \geq d \cdot \epsilon\} \cdot \frac{\tilde{z}^{d-k+1} e^{-\tilde{z}}}{\Gamma(d-k)} \cdot e^{-\epsilon} d\epsilon d\tilde{z} \\
&= \int_0^\infty \left(1 - e^{-(f_k(z_{1:k}) + \tilde{z})/d}\right) \cdot \frac{\tilde{z}^{d-k+1} e^{-\tilde{z}}}{\Gamma(d-k)} d\tilde{z} \\
&= 1 - e^{-f_k(z_{1:k})/d} \cdot \left(1 + \frac{1}{d}\right)^{k-d},
\end{aligned} \tag{16}
$$

where the first line comes from the definition of the objective vector (15), the second line comes from changing variables that $\tilde{z} = \sum_{i=k+1}^{d} z_i$ and the additivity of Gamma distributions, and others come from the direct calculation. Then, by taking the complement, we have

$$\mathbb{P}(c \leq 0) = e^{-f_k(z_{1:k})/d} \cdot \left(1 + \frac{1}{d}\right)^{k-d}. \tag{17}$$

Next, we compute the probability that $x_\alpha^*(z_{1:k}) = 0$. In the one-dimensional setting with the feasible set $\{x : -1 \leq x \leq 1\}$, the optimal solution of (5) can be determined by the sign of the objective vector. Thus, if we are confident that $c$ is non-negative, the optimal solution should be $x_\alpha^*(z_{1:k}) = -1$; if we are confident that $c$ is non-positive, the optimal solution is $x_\alpha^*(z_{1:k}) = 1$; if we are not confident about the sign of the objective vector, we will be conservative and choose $x_\alpha^*(z_{1:k}) = 0$. That is,

$$x_\alpha^*(z_{1:k}) = \begin{cases} 1, & \text{if } \mathbb{P}(c \leq 0|z_{1;k}) \geq \alpha, \\ 0, & \text{if } \mathbb{P}(c \leq 0|z_{1;k}), \mathbb{P}(c \geq 0|z_{1;k}) \leq \alpha, \\ -1, & \text{if } \mathbb{P}(c \geq 0|z_{1;k}) \geq \alpha. \end{cases} \tag{18}$$

Then, plugging (16) and (17) into (18), we have $\mathbb{P}(c \leq 0|z_{1;k}), \mathbb{P}(c \geq 0|z_{1;k}) \leq \alpha$ only when

$$\max\left\{0, -d\log\left(\alpha\left(1+\frac{1}{d}\right)^{d-k}\right)\right\} \leq f_k(z_{1:k}) \leq \max\left\{0, -d\log\left((1-\alpha)\left(1+\frac{1}{d}\right)^{d-k}\right)\right\}. \tag{19}$$

Integrating the probability on the set (19) with respect to $z_{1:k}$, we have

$$\mathbb{P}\left(x_\alpha^*(z_{1:k}) = 0\right) = \tag{20}$$

$$\frac{1}{\Gamma(k)}\left(\gamma\left(k, \max\left\{0, -d\log\left((1-\alpha)\left(1+\frac{1}{d}\right)^{d-k}\right)\right\}\right) - \gamma\left(k, \max\left\{0, -d\log\left(\alpha\left(1+\frac{1}{d}\right)^{d-k}\right)\right\}\right)\right).$$
$$\tag{21}$$

To see it decrease, we can apply Jensen's inequality to see the result. $\qquad\square$

### B.4 Proof for Proposition 3

**Proposition 3.** *For $\alpha \geq 1/2$, let the uncertainty set be*

$$\mathcal{U}_\alpha(z) = \begin{cases} \left[\sqrt{z} - \frac{1-\sqrt{2-2\alpha}}{2}, \infty\right), & \text{if } z \in [0, 1], \\ \left(-\infty, -\sqrt{-z} + \frac{1-\sqrt{2-2\alpha}}{2}\right], & \text{if } z \in [-1, 0). \end{cases}$$

*The uncertainty set has a coverage guarantee in that $\mathbb{P}(c \in \mathcal{U}_\alpha(z)) = \alpha$. If we solve the optimization problem (5) with the uncertainty set $\mathcal{U}_\alpha(z)$, the following robust solution is obtained*

$$x(z) = \begin{cases} -1, & \text{if } z \geq \frac{3-2\alpha-2\sqrt{2-2\alpha}}{4}, \\ 0, & \text{if } \frac{-3+2\alpha+2\sqrt{2-2\alpha}}{4} \leq z \leq \frac{3-2\alpha-2\sqrt{2-2\alpha}}{4}, \\ 1, & \text{if } z \leq \frac{-3+2\alpha+2\sqrt{2-2\alpha}}{4}. \end{cases}$$

*Proof.* We first show that $\mathbb{P}(c \in \mathcal{U}_\alpha(z)) = \alpha$. Given the prediction function $\hat{f}(z) = \text{sign}(z) \cdot \sqrt{|z|}$, the residual is $r = c - \hat{f}(z) = \epsilon\sqrt{|z|}$. By calculation, we have that the marginal distribution of the objective vector $c$ satisfies

$$\mathbb{P}(r \leq r_0) = \begin{cases} -2r_0^2 + 2r_0 + \frac{1}{2}, & \text{if } 0 \leq r_0 \leq 1/2, \\ 2r_0^2 + 2r_0 + \frac{1}{2}, & \text{if } -1/2 \leq r_0 \leq 0, \end{cases} \tag{22}$$

and this distribution is the same as the marginal distribution of the objective vector given $z \geq 0$ or $z \leq 0$ since the distribution of $r$ only depends on the absolute value of $z$. Then, we have

$$\mathbb{P}(c \in \mathcal{U}_\alpha(z)) = \frac{1}{2}\mathbb{P}\left(r \geq -\frac{1-\sqrt{2-2\alpha}}{2}\,\middle|\, z \geq 0\right) + \frac{1}{2}\mathbb{P}\left(r \leq \frac{1-\sqrt{2-2\alpha}}{2}\,\middle|\, z < 0\right)$$

$$= \frac{1}{2}\mathbb{P}\left(r \geq -\frac{1-\sqrt{2-2\alpha}}{2}\right) + \frac{1}{2}\mathbb{P}\left(r \leq \frac{1-\sqrt{2-2\alpha}}{2}\right)$$

$$= \alpha,$$

where the first line comes from the property of the conditional expectation, the second line comes from the fact that the distribution of the residual is independent of the sign of the covariate $z$, and the last line comes from (22).

Next, we show that the solution $x(z)$ is the optimal solution of problem (5) with the uncertainty set $\mathcal{U}_\alpha(z)$. By (5) and the feasible set, we have that the optimal solution is 1 or $-1$ only when all elements in the uncertainty set is non-positive or non-negative, respectively, and the optimal solution is 0 when the uncertainty set contains both positive and negative numbers. Thus, we can find the result by computing the lower and upper bound of the uncertainty set $\mathcal{U}_\alpha(z)$ for $z \in [0, 1]$ and $z \in [-1, 0)$, respectively. $\qquad\square$

## C  Supplementary for Section 4

This section provides our algorithm and the corresponding analysis for Section 4. We first briefly summarize the problem and then, develop the DRO algorithm raised by our predict-then-calibrate framework and the detailed proof of the algorithm analysis.

As stated in Section 4, we here consider the risk-neutral problem (2) as follows

$$\min_{x} \mathbb{E}[c|z]^\top x,$$
$$\text{s.t. } Ax = b, \ x \geq 0,$$

instead of a risk-sensitive objective in Section 3. As stated in Section 2, we work on a well-trained ML prediction model $\hat{f}$ that predicts the objective vector $c$ based on the covariate $z$. Then, we will apply the idea of distributionally robust optimization to bound the prediction error and develop the generalization bound. We remark again that the prediction model can be any off-the-shelf machine learning method since we have no assumption about it.

## C.1 DRO Algorithm

In this part, we derive the DRO algorithm from our predict-them-calibrate framework. As for Algorithms 1 and 2, we work on the prediction error

$$r_t = c_t - \hat{f}(z_t)$$

of all samples in the validation dataset $\mathcal{D}_{val} = \{c_t, A_t, b_t, z_t\}_{t=1}^T$. In the following, by a little abuse of notation, we use $t \in \mathcal{D}_{val}$ to denote a sample tuple $(c_t, A_t, b_t, z_t)$ in the validation set.

---

**Algorithm 4** Contextual Distributionally Robust LP

1: Input: Dataset $\mathcal{D}_{val}$, ML model $\hat{f}$, radius $\varepsilon$, bandwidth $h$, target covariate $z_0$
2: For each $t \in \mathcal{D}_{val}$, let
$$r_t := c_t - f(z_t)$$
3: Define the estimation of the conditional mean of residuals of covariate $z_0$ by
$$r_0 := \sum_{t \in \mathcal{D}_{val}} w_t(z_t, z_0) r_t,$$

where

$$w_t(z) \propto K((z_t - z_0)/h), \ \sum_{t \in \mathcal{D}_{val}} w_t(z) = 1$$

and $K(z)$ is a kernel function that satisfies Assumption 3
4: Construct the degenerate ambiguity set by
$$\Xi = \{r' : \|r' - r_0\|_2 \leq \varepsilon\}$$

5: Solve the degenerate distributional robust optimization problem
$$\hat{x}(z_0) = \arg\min_{x} \ \sup_{r \in \Xi}(f(z) + r)^\top x,$$
$$\text{s.t. } Ax = b, \ x \geq 0,$$

6: Output: $\hat{x}(z_0)$

---

The main idea of our algorithm is similar to distributionally robust optimization that consists of two steps: firstly, construct an ambiguity set $\Xi$, and secondly, solve the following distributionally robust LP:

$$\min_{x} \max_{\mathcal{P}' \in \Xi} \mathbb{E}_{\mathcal{P}'}\left[(\hat{f}(z) + r)^\top x\right], \tag{23}$$
$$\text{s.t. } Ax \leq b, \ x \geq 0,$$

and use its optimal solution to solve (2). If the ambiguity set contains the target distribution, say $\mathcal{P}_{r|z_0}$, we can obtain a good solution to (2) by solving (23) with some generalization bound. Here, $\mathcal{P}_{r|z}$ denotes the distribution of the prediction error given the covariate $z$. In our case, specifically, the ambiguity set can degenerate into a set consisting of the mean of distributions in the ambiguity set since we only focus on the estimation of the conditional mean $\mathbb{E}[r|z_0]$ for the target covariate $z_0$. Thus, in the following, we will also use $\Xi$ to denote a set of vectors corresponding to the means of

the distributions in $\Xi$. Then, as the ambiguity set degenerates, the problem (23) also degenerates to the robust optimization problem in Step 5 of Algorithm 4.

Now, we consider the construction of $\Xi$. We apply a kernel estimation in Steps 3 and 4 of Algorithm 4 to construct $\Xi$. The following proposition says that if we choose proper $\varepsilon$ in Algorithm 4, $\Xi$ contains the conditional mean $\mathbb{E}[r|z_0]$.

**Proposition 6.** *Assume there exists a constant $L$ such that the following condition holds for any $z, z'$*

$$|\mathbb{E}[r|z] - \mathbb{E}[r|z']| \leq L\|z - z'\|_2^s.$$

*Then under mild boundedness assumptions on the distribution, with probability no less than $1 - \delta$, $r_0$ defined in Algorithm 4 satisfies*

$$\|\mathbb{E}[r|z_0] - r_0\|_2 \leq O\left(\left(\frac{Ld^{s/2}}{\delta} + \frac{32}{\delta^{1/2}}\right) \cdot T^{-\frac{s}{2(s+d)}} \log T\right)$$

*when choosing the bandwidth $h = T^{-\frac{1}{2s+2d}}$.*

We defer the proof and assumptions to the next chapter. After developing this estimation error, if the feasible region of (1) is in the unit ball with probability 1, we can directly build the generalization bound for the downstream problem by Proposition 4:

**Proposition 4.** *Under the same assumptions as in Proposition 6, Algorithm 4 outputs a solution $\hat{x}(z_0)$ such that*

$$\mathbb{E}\left[c^\top (\hat{x}(z_0) - x^*(z_0))\right] \leq O\left(\left(\frac{Ld^{s/2}}{\delta} + \frac{32}{\delta^{1/2}}\right) \cdot T^{-\frac{s}{2(s+d)}} \log T\right).$$

*holds with probability no less than $1 - \delta$.*

Here, we emphasize that if the residuals are i.i.d., we can equivalently have $L = 0, s = \infty$. Then, the convergence rate of our Algorithm 4 is $O(T^{-1/2})$, which matches the rate in Wang et al. [2021], Kannan et al. [2020, 2021]. Moreover, we remark that our results have the potential to be generalized to general cases for distributionally robust optimization if there are more convergence theories developed for the non-i.i.d. samples. Specifically, if the distributions have some smooth properties with respect to the covariate $z$, the kernel estimation

$$\mathcal{P}'_{r|z} = \sum_{t \in \mathcal{D}_{val}} w_t(z_t, z)\mathcal{P}_{r|z_t}, \tag{24}$$

will also be a good approximation to $\mathcal{P}_{r|z_0}$, where $w_t$ is the weight constructed in Step 3 in Algorithm 4 with some kernel function for all $t \in \mathcal{D}_{val}$. The proof will be similar to the proof of Proposition 6. Once being able to utilize samples in $\mathcal{D}_{val}$ to approximate $\mathcal{P}'_{r|z}$, we can develop a similar DRO algorithm for general cases. However, to our best knowledge, the current state-of-the-art theory can only approximate a distribution by the empirical distribution constructed by i.i.d. samples [Fournier and Guillin, 2015], and its proof cannot be extended to a more general non-i.i.d setting. Thus, one interesting future direction will be developing distributional convergence results for non-i.i.d. settings, and our framework can also have the potential to be applied in general distributionally robust problems.

In the following sections, we will provide detailed proof for Proposition 6 and 4. We first state and interpret our assumptions in Section C.2. Then, in Section C.3, we show two essential lemmas that are helpful for later proof. In Sections C.4 and C.5, we will rigorously restate and show those two propositions, respectively.

## C.2 Assumptions for Algorithm 4

Now, we state those assumptions. Most of them are boundedness assumptions. We first state our assumptions about the residual. Recall the residual defined in Algorithm 4 is

$$r = c - \hat{f}(z),$$

for any obejective vector $c$ and covariate $z$, where $\hat{f}$ is the prediction model.

**Assumption 1.** *We assume the residual vector $r$ satisfies the following conditions:*

(a) *Boundedness: the residual vector is bounded that $r \in [-1, 1]^m$.*

(b) *Distribution: the residual vector $r$ given the covariate $z$ follows some unknown continuous distribution $\mathcal{P}_{r|z}$ with the density function $p_{r|z}$.*

(c) *Smoothness: for any two covariates $z, z' \in [0, 1]^d$, the difference between the conditional means of $\mathcal{P}_{r|z}$ and $\mathcal{P}_{r|z'}$ satisfies*

$$\|\mathbb{E}[r|z] - \mathbb{E}[r|z']\|_2 \leq L \|z - z'\|_2^s$$

*for some positive constants $L, s > 0$.*

Assumptions 1-(a),(b) are just boundedness conditions. Assumption 1-(c) says that the distribution of the residual vector $r$ given the covariate $z$ enjoys some smoothness, and $s$ measures the order of the smoothness. Specifically, if we perturb the covariate $z$ a little bit, the perturbation will only slightly change the corresponding conditional distribution of the residual $r$ given $z$. This condition is weaker than many recent papers with some true-model assumptions. For example, Wang et al. [2021], Kannan et al. [2020] assume $c = f_0(z) + \epsilon$ for some function $f_0(z)$ and random noise $\epsilon$, and Kannan et al. [2021] also assume that all objective vectors for different covariates share a same type of randomness. When $f_0(z)$ can be learned by the prediction model, Assumption 1-(c) holds for $L = 0$ and $s = \infty$.

**Assumption 2.** *We assume the covariate $z$ satisfies the following conditions:*

(a) *Boundedness: the covariate is bounded that $z \in [0, 1]^d$.*

(b) *Distribution: the covariate $z$ follows some unknown continuous distribution $\mathcal{P}_z$ with the density function $p(z) \in C^1([0, 1]^d)$.*

(c) *Smoothness: the density function $p(z) \in C^1([0, 1]^d)$ satisfies $p(z) \leq \bar{p}$ and $\|\nabla p(z)\|_2 \leq c$ for some positive constants $\bar{p}, c \geq 1$ and all $z \in [0, 1]^d$.*

The first two parts of Assumption 2 are also boundedness assumptions. Part (c) guarantees that the density function is smooth enough so that it has no steep peak and can not change drastically around any covariate $z$. We remark that we use this assumption only for simplicity, and it can be replaced by other weaker assumptions that can be satisfied by any density function. The last assumption is about the kernel function we used in Algorithm 4. It basically requires the kernel function to have bounded support and a positive lower bound on the support. This condition can also be satisfied by many kernel functions, such as the uniform kernel and the truncated Gaussian kernel.

**Assumption 3.** *We assume the kernel function satisfies*

$$K(z) \geq b_r \cdot \mathbb{1}\{K(z) > 0\} \text{ and } K(z) \leq b_R \cdot \mathbb{1}\left\{ \max_{i=1,\dots,d} |z_i| \leq R \right\},$$

*for some positive constants $b_R, b_r > 0$ and $R \geq 1$.*

### C.3 Essential Lemmas of Proving Propositions 6 and 4

In this section, we let $T = |\mathcal{D}_{val}|$ and the bandwidth $h = T^{-\gamma}$, where $\gamma$ is a parameter that we will choose later. Without loss of generality, we let $\mathcal{D}_{val} = \{1, ..., T\}$. For any vector $z \in \mathbb{R}^d$, we use $(z)_i$ to denote its $i$-th entry for $i = 1, ..., d$. In addition, denote $z_0$ as the target covariate, denote $K_t = K(\frac{z_t - z_0}{h})$ as the value of the kernel function evaluated at $z_t$ with bandwidth $h$, and denote $w_t(z_t) = \frac{K(\frac{z_t - z_0}{h})}{\sum_{i=1}^T K(\frac{z_i - z_0}{h})}$ as the corresponding weight for any $t \in \mathcal{D}_{val}$. Moreover, for simplicity, by a little abuse of notation, we drop the input variables of the kernel and weight functions and use $K_t$ and $w_t$ to represent the value of the kernel function and the weight when the context is clear.

Before proving Propositions 6 and 4, we first provide two essential lemmas. Lemma 1 shows that, with high probability, there are at least $O(T^{1-d\gamma})$ samples with positive weights. In the proof of the later lemma and Propositions 6 and 4, we will utilize those samples to give an approximation of the conditional mean of the target distribution $\mathcal{P}_{r|z}$.

**Lemma 1.** *Under Assumption 2, for any positive constant $\delta \in (0,1)$ and $T \geq \max\left\{10, \left(\frac{\delta}{8c}\right)^{-\frac{1}{\gamma}}\right\}$, with probability no less than $1 - \frac{1}{T^2} - \frac{\delta}{2}$,*

$$\sum_{t=1}^{T} 1\{w_t > 0\} \geq \frac{\delta}{4}(2R)^d T^{1-d\gamma} - \sqrt{T}\log T. \tag{25}$$

*Specifically, if $1 - d\gamma > \frac{1}{2}$ and $T$ is sufficiently large such that $\frac{\delta}{2}(2R)^d T^{1-d\gamma} \geq 2\sqrt{T}\log T$,*

$$\sum_{t=1}^{T} 1\{w_t > 0\} \geq \frac{\delta}{8}(2R)^d T^{1-d\gamma}. \tag{26}$$

*Proof.* In the following, we show the first inequality (25). The second inequility (26) can be obtained by plugging the condition $\frac{\delta}{8}(2R)^d T^{1-d\gamma} \geq 2\sqrt{T}\log T$ into (25).

By Hoeffding's inequality, we have for any target covariate $z_0$, with probability no less than $1 - \frac{1}{T^2}$,

$$\sum_{t=1}^{T} 1\{w_t > 0\} \geq T\mathbb{E}\left[1\left\{\max_{i=1,\ldots,d}|(z)_i - (z_0)_i| \leq R \cdot T^{-\gamma}\right\}\right] - \sqrt{T}\cdot\log T,$$

where the probability is taken with respect to $z_t$ for $t = 1, \ldots, T$. Thus, it is sufficient to show that with probability no less than $1 - \frac{\delta}{2}$

$$\mathbb{E}\left[1\left\{\max_{i=1,\ldots,d}|(z)_i - (z_0)_i| \leq R \cdot T^{-\gamma}\right\}\right] \geq \frac{\delta}{4}(2R)^d T^{-d\gamma},$$

where the probability is taken with respect to $z_0$. To see this, for any $z_0$ such that $p(z_0) > \delta/4$, by Assumption 2-(c), we have that when $T \geq \left(\frac{4\sqrt{d}c}{\delta}\right)^{1/\gamma}$,

$$p(z) \geq p(z_0) - c\sqrt{d}R \cdot T^{-\gamma}$$
$$\geq \frac{\delta}{2} - \frac{\delta}{4} = \frac{\delta}{4}$$

for all $z$ satisfying $\max_{i=1,\ldots,d}|(z)_i - (z_0)_i| \leq R\cdot T^{-\gamma}$, where the first line comes from Taylor's expansion and Assumption 2-(c), and the second line comes the condition that $T \geq \left(\frac{4\sqrt{d}c}{\delta}\right)^{1/\gamma}$. Thus, by integrating the density function on the set $\left\{z : \max_{i=1,\ldots,d}|(z)_i - (z_0)_i| \leq R \cdot T^{-\gamma}\right\}$, we have

$$\mathbb{E}\left[1\left\{\max_{i=1,\ldots,d}|(z)_i - (z_0)_i| \leq R \cdot T^{-\gamma}\right\}\right] \geq \frac{\delta}{4}(2R)^d T^{-d\gamma}.$$

Moreover, since $\mathbb{P}(\{z : p(z) < \delta/2\}) = \int p(z)1\{(p(z) < \delta/2)\}\mathrm{d}z \leq \delta/2$, we have $\mathbb{P}(\{z : p(z) \geq \delta/2\}) \geq 1 - \delta/2$, and we finish the proof. $\square$

The next lemma develops the concentration property for $r_0$ defined in Step 3 in Algorithm 4.

**Lemma 2.** *Under Assumptions 1 and 3, for any $T > 0$, with probability no less than $1 - \frac{2n}{T^2}$, the inequality below holds for all $i = 1, \ldots, n$*

$$\left|\sum_{t=1}^{T} w_t(r_t)_i - \sum_{t=1}^{T} w_t\mathbb{E}[(r_t)_i|z_t]\right| \leq \frac{2b_R}{b_r}\cdot\tilde{T}^{-\frac{1}{2}}\cdot\log T,$$

*where $\tilde{T} = \sum_{t=1}^{T} 1\{w_t > 0\}$. Here, the probability is taken with respect to the covariates $z_1, \ldots, z_T$ and their corresponding residuals.*

*Proof.* This is also an application of Hoeffding's inequality. We first show it for fixed $z_1, ..., z_T$ and fixed target covariate $z_0$. In this case, we have $\tilde{T} = \sum_{t=1}^{T} 1\{w_t > 0\}$ is fixed as well. Moreover, for any $w_t > 0$, by Assumption 3 and its definition in Algorithm 4 that it is a ratio between corresponding non-zero kernel functions, we have $w_t \in \left[\frac{b_r}{b_R}, \frac{b_R}{b_r}\right]$. In addition, by Assumption 1-(a) that each entry of the residual is in $[-1, 1]$, we have $w_t \cdot (r_t)_i$ is bounded by $[-\frac{b_R}{b_r}, \frac{b_R}{b_r}]$. Thus, by Hoeffding's inequality (Lemma 3), we have with probability no less than $1 - 2/T^2$,

$$\left|\sum_{t=1}^{T} w_t \cdot (r_t)_i - \sum_{t=1}^{T} w_t \mathbb{E}[(r_t)_i | z_t]\right| \leq \frac{2b_R}{b_r} \cdot \tilde{T}^{-\frac{1}{2}} \cdot \log T,$$

for any $i = 1, ..., d$. Then, we can find the result by taking the union bound for all entries $i = 1, ..., n$ and integrating the probability with respect to $z_1, ..., z_T$. □

Now, we are equipped with all essential lemmas, and in the following, we will show Propositions 6 and 4.

## C.4 Proof of Proposition 6

In this section, we will adapt the same notation as in Section C.3.

**Proposition 6.** *Under Assumptions 1, 2 and 3, with probability no less than $1 - \delta$,*

$$\|\mathbb{E}[r|z_0] - r_0\|_2 \leq \frac{b_R}{b_r} \cdot \left(\frac{16L}{\delta} \cdot \bar{p}d^{s/2}R^s + \frac{32\sqrt{n}}{\delta^{1/2}}\right) \cdot T^{-\frac{s}{2s+2d}} \log T$$

*holds when choosing the bandwidth $h = T^{-\frac{1}{2s+2d}}$.*

*Proof.* To prove it, we first apply the triangle inequality that

$$\|\mathbb{E}[r|z_0] - r_0\|_2 \leq \|\mathbb{E}[r|z_0] - \mathbb{E}[r_0|z_0, ..., z_T]\|_2 + \|r_0 - \mathbb{E}[r_0|z_0, ..., z_T]\|_2$$

$$= \|\mathbb{E}[r|z_0] - \sum_{t=1}^{T} w_t \mathbb{E}[r|z_t]\|_2 + \|r_0 - \mathbb{E}[r_0|z_0, ..., z_T]\|_2 \tag{27}$$

$$\leq L \cdot \sum_{t=1}^{T} w_t \|\mathbb{E}[r|z_0] - \mathbb{E}[r|z_t]\|_2 + \|r_0 - \mathbb{E}[r_0|z_0, ..., z_T]\|_2$$

$$\leq L \cdot \sum_{t=1}^{T} w_t \|z_0 - z_t\|_2^s + \|r_0 - \mathbb{E}[r_0|z_0, ..., z_T]\|_2$$

where the first and the third line come from the triangle inequality, the second line comes from the definition of $r_0 = \sum_{t=1}^{T} w_t r_t$, and the last line comes from Assumption 1-(c). In the following, we analyze the first and the second terms in the last line, respectively.

We first analyze the first term in the last line of (27). Denote $\tilde{T} = \sum_{t=1}^{T} 1\{w_t > 0\}$. By Hoeffding's inequality, for fixed $z_0$, we have with probability no less than $1 - \frac{2}{T^2}$,

$$\sum_{t=1}^{T} w_t \|z_0 - z_t\|_2^s = \frac{1}{\sum_{t=1}^{T} K_t} \sum_{t=1}^{T} K_t \|z_0 - z_t\|_2^s$$

$$\leq \frac{1}{b_r \tilde{T}} \sum_{t=1}^{T} K_t \|z_0 - z_t\|_2^s \tag{28}$$

$$\leq \frac{1}{b_r \tilde{T}} \left(T\mathbb{E}[K_1\|z_1 - z_0\|_2^s] + b_R d^{s/2} R^s T^{1/2 - s\gamma} \log T\right),$$

where the first line comes from the definition of $w_t$, the second line comes from Assumption 3, which gives the lower bound of the kernel function, and the third line comes from Hoeffding's inequality. Then, we compute the expectation

$$
\begin{aligned}
\mathbb{E}[K_1 \| z_1 - z_0 \|_2^s] &\leq \mathbb{P}(K_1 > 0) \cdot b_R d^{s/2} R^s T^{-s\gamma} \\
&\leq \bar{p} \cdot (2RT^{-\gamma})^d \cdot b_R d^{s/2} R^s T^{-s\gamma} \\
&= \bar{p} b_R 2^d R^{d+s} T^{-s\gamma - d\gamma}.
\end{aligned}
$$

Here, the first step is obtained by Assumption 3, the upper bound of the kernel function and the boundedness of its support, the second line comes from Assumption 2, the upper bound of the density function, and the last line is obtained by calculation. Plugging this upper bound into (28), we have the first term in the right-hand side of (27) can be bounded by

$$
L \sum_{t=1}^{T} w_t \| z_0 - z_t \|_2^s \leq \frac{L b_R}{b_r \tilde{T}} \left( 2^d \bar{p} R^{d+s} T^{1-s\gamma-d\gamma} + d^{s/2} R^s T^{1/2-s\gamma} \log T \right). \tag{29}
$$

Then, for the second term in the right-hand side of (27), under the event of Lemma 2, we have for fixed $z_0$

$$
\| r_0 - \mathbb{E}[r_0 | z_0, ..., z_T] \|_2 \leq \frac{2 b_R \sqrt{n}}{b_r} \cdot \tilde{T}^{-\frac{1}{2}} \cdot \log T. \tag{30}
$$

Next, we combine (29) and (30) to draw the result. Under the event of Lemma 1 that

$$
\tilde{T} \geq \frac{\delta}{8} (2R)^d T^{1-d\gamma},
$$

the inequality (29) becomes

$$
L \sum_{t=1}^{T} w_t \| z_0 - z_t \|_2^s \leq \frac{8 L b_R}{\delta b_r} \left( \bar{p} R^s + d^{s/2} 2^{-d} R^{s-d} \right) \cdot T^{- \min\{s\gamma, 1/2+s\gamma-d\gamma\}} \log T, \tag{31}
$$

and the inequality (30) becomes

$$
\| r_0 - \mathbb{E}[r_0 | z_0, ..., z_T] \|_2 \leq \frac{b_R \sqrt{n}}{b_r 2^{(d-5)/2} R^{d/2} \delta^{1/2}} \cdot T^{-\frac{1-d\gamma}{2}} \cdot \log T. \tag{32}
$$

The event of the intersection of events corresponding to Lemma 1 and inequalities (29) and (30) happends with probability no less than $1 - \frac{\delta}{2} - \frac{2n+3}{T^2}$. Thus, if $T \geq \sqrt{\frac{4n+6}{\delta}}$, plugging (31) and (32) into (27), we have with probability no less than $1 - \delta$,

$$
\| \mathbb{E}[r | z_0] - r_0 \|_2 \leq \frac{b_R}{b_r} \cdot \left( \frac{16L}{\delta} \cdot \bar{p} d^{s/2} R^s + \frac{32\sqrt{n}}{\delta^{1/2}} \right) \cdot T^{- \min\{s\gamma, 1/2+s\gamma-d\gamma, 1/2-d\gamma/2\}} \log T.
$$

Finally, let $\gamma = \frac{1}{2s+2d}$, we finish the proof.

$\qquad\qquad\qquad\qquad\qquad\qquad\qquad\qquad\qquad\qquad\qquad\qquad\qquad\qquad\qquad\qquad\qquad\qquad\qquad\quad$ $\square$

## C.5 Proof of Proposition 4

In this section, we also use the same notation as in Sections C.2 and C.4. We show that the optimality gap between the solution given by Algorithm 4 and the optimal solution converges to 0 as $T \to \infty$.

**Proposition 4.** *Suppose the feasible region $\{x : Ax \leq b, x \geq 0\}$ is included in the unit ball $\{x : \|x\| \leq 1\}$. and the absolute values of each entry of the prediction function is bounded by $\bar{f}$. Under Assumptions 1, 2 and 3, Algorithm 4 outputs a solution $\hat{x}(z_0)$ such that with probability no less than $1 - \delta$*

$$
\mathbb{E}\left[ c^\top \left( \hat{x}(z_0) - x^*(z_0) \right) \big| z_0 \right] \leq \frac{b_R}{b_r} \left( \frac{32L}{\delta} \cdot \bar{p} d^{s/2} R^s + \frac{64}{\delta^{1/2}} \right) \cdot T^{-\frac{s}{2s+2d}} \log T,
$$

*where $x^*(z_0)$ is the optimal solution to $LP(\mathbb{E}[c|z_0], A, b)$, and the probability is taken with respect to $z_1, ..., z_T$, their corresponding objective vectors, the constraints $(A, b)$ and the target covariate $z_0$.*

*Proof.* This is a direct application of Proposition 6. We first show the relation between two LPs with different objective vectors. For any two LPs with different objective vectors $c_1, c_2$ but with the same constraint $(A, b)$, denote $x_1^*$ and $x_2^*$ as their optimal solutions. Then, if the optimal solutions are contained in the unit ball, we have

$$
\begin{aligned}
c_1^\top (x_2^* - x_1^*) &= \left( c_1^\top x_2^* - c_2^\top x_2^* \right) + \left( c_2^\top x_2^* - c_2^\top x_1^* \right) + \left( c_2^\top x_1^* - c_1^\top x_1^* \right) \\
&\leq \|c_1 - c_2\|_2 \|x_1^*\|_2 + \|c_1 - c_2\|_2 \|x_1^*\|_2 + \left( c_2^\top x_2^* - c_2^\top x_1^* \right) \\
&\leq \|c_1 - c_2\|_2 \|x_1^*\|_2 + \|c_1 - c_2\|_2 \|x_1^*\|_2 \\
&\leq 2\|c_1 - c_2\|_2,
\end{aligned}
\tag{33}
$$

where the first step is obtained by rearranging terms, the second step is obtained by Cauchy inequality, the third step is obtained by the optimality of $x_2^*$ with respect to the objective vector $c_2$, and the last step is obtained by the condition that $x_1$ and $x_1$ are in the unit ball.

Then, since in the event of Proposition 6 that happens with probability no less than $1 - \delta$,

$$
\|\mathbb{E}[r|z_0] - r_0\|_2 \leq \frac{b_R}{b_r} \cdot \left( \frac{16L}{\delta} \cdot \bar{p} d^{s/2} R^s + \frac{32}{\delta^{1/2}} \right) \cdot T^{-\frac{s}{2s+2d}} \log T,
$$

and the difference between the objective vectors is the same as the difference between the corresponding residuals, the inequality (33) implies that

$$
\mathbb{E}[c|z_0]^\top \left( x^*(z_0) - \hat{x}(z_0) \right) \leq \frac{b_R}{b_r} \left( \frac{32L}{\delta} \cdot \bar{p} d^{s/2} R^s + \frac{64}{\delta^{1/2}} \right) \cdot T^{-\frac{s}{2s+2d}} \log T,
\tag{34}
$$

and we finish the proof. $\qquad\square$

## D    Auxiliary Lemmas

In this section, we state some useful lemmas in the information theory that are helpful in our proof.

**Lemma 3** (Hoeffding's inequality). *Let $X_1, ..., X_T$ be independent random variables such that $X_t$ takes its values in $[u_t, v_t]$ almost surely for all $t \leq T$. Then the following inequality holds*

$$
\mathbb{P}\left( \left| \frac{1}{T} \sum_{t=1}^n X_t - \mathbb{E}X_t \right| \geq s \right) \leq 2 \exp\left( -\frac{2T^2 s^2}{\sum_{i=1}^n (u_t - v_t)^2} \right)
$$

*for any $s > 0$.*

*Proof.* We refer to Chapter 2 of Boucheron et al. [2013]. $\qquad\square$

## E    Additional Related Work

Due to the space constraints in the main text, we provide additional discussion on related work here.

**Contextual RO with a parametrized prediction model.** An alternative approach to contextual robust optimization relies on the assumption that the relationship between the outcome variable $c$ and the covariate $z$ is governed by a parameter $\theta$ in the true underlying model. In this approach, the estimation/prediction process and the optimization problem are integrated by constructing an uncertainty set for the parameter $\theta$ to mitigate the estimation uncertainty. Zhu et al. [2022] constructs the uncertainty set for $\theta$ that contains all the parameters with training loss no worse than a threshold, but no coverage guarantee is provided. This can also be viewed as a special case of the decision-driven regularization model introduced in Loke et al. [2022]. Another study by Cao and Gao [2021] extends the consideration of uncertainty to both the parameter and residual. However, the incorporation of estimation and optimization processes in one optimization model can raise tractability issues, restricting the choice of complex yet expressive prediction models.

For the contextual stochastic optimization problem with a risk-neutral objective, below is the other two streams of work in addition to the predict-then-optimize paradigm illustrated in the main text.

**Local learning-based conditional stochastic optimization.** One stream of work tends to construct stochastic optimization problems with estimated conditional distributions learned by some local

learning methods. Hannah et al. [2010] approximates it by using Nadaraya-Watson kernel regression [Nadaraya, 1964, Watson, 1964] to reweight the historical data. Bertsimas and Kallus [2020] further generalizes it to a framework using a broader range of non-parametric machine learning methods based on local learning, such as $k$NN [Altman, 1992], classification and regression trees [Breiman, 2017]. To overcome the overfitting effect of these methods when the sample size is small, distributionally robust optimization (DRO) methods are introduced. Bertsimas and Van Parys [2022] enhances the robustness of bootstrap data by constructing the entropy-based DRO model. Nguyen et al. [2020, 2021], Wang et al. [2021], Bertsimas et al. [2023] investigate the Wasserstein-based DRO model. Ho and Hanasusanto [2019] designs a regularized approach and obtains a tractable approximation by leveraging ideas from DRO with a modified $\chi^2$ ambiguity set.

**End-to-end prescriptive analytics without prediction.** Instead of learning to predict uncertain outcomes, Ban and Rudin [2019] directly learns the function mapping from covariates to decisions by employing the linear decision rule for the newsvendor problem. There are also further studies investigating such integrated approaches without prediction using more complicated models [Bertsimas and Koduri, 2022], such as random forests [Kallus and Mao, 2022] and neural networks [Chen et al., 2022]. Nevertheless, the robustness of the decisions to the epistemic and aleatoric uncertainty is rarely considered in these papers.

