# OpenReview forum: "Predict-then-Calibrate: A New Perspective of Robust Contextual LP"
_NeurIPS.cc/2023/Conference — NeurIPS 2023 poster_

### Official Review · Reviewer_tVPZ · 2023-06-25

**Soundness:** 3 good
**Presentation:** 2 fair
**Contribution:** 2 fair
**Rating:** 4
**Confidence:** 3

**Summary:**

The authors study a risk-averse variant of contextual linear optimization, using VaR as the risk measure. The authors develop two heuristic approaches. In both of these approaches, the problem of minimizing the VaR is approximated by a robust optimization problem. The authors propose two different ways of specifying the uncertainty sets, both of which involve building a regression model to predict the conditional mean of the uncertainty, and then calibrating the size of uncertainty with the intent of satisfying a coverage guarantee. The authors provide bounds on the probability that the coverage guarantee will be satisfied, and computational experiments that measure the performance of the proposed method against several baselines. In addition, the authors briefly discuss a distributionally robust variant of contextual linear optimization, and show that there exists a policy that converges to an optimal policy with a specific convergence rate.

**Strengths:**

- The problem selected by the authors, namely risk-sensitive contextual linear optimization, is an interesting and relevant problem.
- The design of the experiments provided is reasonable and the proposed methods show good performance.

**Weaknesses:**

- The exact form of the uncertainty set suggested by the authors seems poorly motivated. It would be helpful if the authors could explain why they chose the approach that they did. In the rebuttal, the authors provided an explanation that helped to motivate the form of the uncertainty set.

- The coverage guarantees provided by the authors' methods are not the relevant coverage guarantees for the contextual problem (4). In particular, the conditional probability P(c ϵ U_α(z) | z) should be equal to α (approximately), but the authors instead have designed the methods to guarantee that P(c ϵ U_α(z)) is equal to α (approximately). I view this as a serious flaw in the method, as someone solving a contextual linear optimization problem is presumably specifying their risk for that context, and the authors' approach could in theory severely underestimate the risk in heteroskedastic settings. The authors should redesign the method to ensure that the correct coverage guarantee is met. In the rebuttal, the authors explained how the method could be adapted to provide a conditional coverage guarantee, and explained their rationale for using a global coverage guarantee rather than a conditional one.

- Even if the goal were to produce an unconditional coverage guarantee rather than a conditional one, the theoretical guarantees in Proposition 1 and Corollary 1 are weak. Some very bad ways of choosing uncertainty sets would satisfy these guarantees. For example, you could specify your uncertainty set as a ball with a center of 0 and with radius chosen so that the proportion of samples from the validation set falling in the ball is equal to α, and it would have the same guarantees. It would be better if the authors could show some stronger properties that their choice of uncertainty set satisfies. The authors address this in their rebuttal by acknowledging that the theoretical guarantees could be satisfied by bad choices of uncertainty sets, there is other evidence in the paper that suggests that their choice of uncertainty set is effective.

- In Proposition 3, the authors should ensure that "P(c ϵ U_α(z) | z) = α", not that "P(c ϵ U_α(z)) = α" since that is the relevant coverage guarantee for the contextual problem. I considered the possibility that this is a typo, but it can be confirmed that the coverage guarantee for the contextual problem is not satisfied. For example, when z=1, the probability P(c ϵ U_α(z) | z) = 1 - sqrt(2-2α)/2, which is not generally equal to α. This example should be reworked. This was also addressed by the author's discussion of the conditional and global coverage guarantees in the rebuttal.

- Section 4 feels like a summary of a different paper that was inserted into this paper. The proposed method for the contextual DRO problem seems only loosely connected to the methods for the risk-sensitive contextual linear optimization. Furthermore, the authors relegate their proposed method for contextual DRO problems to the appendix and provide no computational results for this method. I recommend removing Section 4. The authors addressed this in their rebuttal by clarifying the connections between the contextual DRO problem and the risk-sensitive problem.

- (A minor point) The authors are inconsistent as to whether the values A and b are random or constant. The methodology seems to assume that these are either fixed or, at the very least, independent of the values of c and z, so I would recommend treating these as constant throughout.

**Questions:**

- In the box uncertainty quantification method, why did the authors choose to perform conditional quantile estimation in two steps, where first the mean is estimated and then quantiles are estimated from the residuals? I would note that in existing works where the mean is estimated and then a density is fit on the residuals, there is usually some assumption that justifies this. For example, if it is assumed that c = f(x) + ε, where the errors ε for different observations are i.i.d., then you can perform unconditional density estimation on the residuals (rather than conditional density estimation). However, I don't see any justification in this work. The authors answered this question in their rebuttal.

**Limitations:**

- One limitation (mentioned in the "Weaknesses" section) is that the coverage guarantees provided by the method are not the ones that would be desired in the stated problem. This is mentioned in passing, but the authors should investigate this limitation further.

- Even in the absence of uncertainty, the proposed method by the authors is a heuristic in the sense that it does not identify the optimal solution nor are any approximation ratios known. It would be worth exploring how close (or far) the solutions are from optimal solutions.

---

> ### Author Rebuttal · Authors · 2023-08-10
>
> We thank the reviewer for the detailed comments and the raised questions. We believe the clarification of these questions will make the positioning of our work clearer.
>
> Individual/conditional coverage guarantee:
>
> We thank the reviewer for noting the point, and we have also mentioned in our paper this as a limitation of the proposed algorithms. However, we note that the predict-then-calibrate (PTC) framework is more of a general framework to tailor a prediction-aimed ML model for a robust task. The performance guarantee of the current two algorithms only hold for global coverage but not for individual coverage, yet this is NOT caused by the framework, but indeed by the chosen calibration/uncertainty quantification algorithm and the corresponding assumption. If we change the calibration algorithm to one that aims for an individual/conditional objective, the whole framework will be able to fulfill the individual coverage. Now we present such an algorithm and the corresponding theory results:
>
> - Calculate the residual $r_t=c_t-\hat{f}(z_t)$
> - Perform a nonparametric regression with respect to $r_t$ using $z_t$ and a chosen kernel $k$ (such as simple window kernel or Gaussian kernel). This obtains the following function
>
> $\hat{Q}(z;\tau) = Q(\sum I(r_t) k(z_t,z)/ \sum k(z_t,z); \tau)$
>
> where  $I(r_t)$ denotes a point mass distribution at $r_t$, and $Q(\cdot;\tau)$ outputs the $\tau$-quantile of a random variable
> - For a new $z_{new}$, output the confidence interval by $[\hat{Q}(z_{new};(1-\alpha)/2), \hat{Q}(z_{new};(1+\alpha)/2)]$
>
> Theoretical analysis:
> Under the Lipschitzness condition of the quantile function
>
> $|Q(c|z;\tau)-\hat{Q}(c|z’;\tau)|\le L\|z-z’\|$
>
> for any $(z,z’)$ and some boundedness assumption, the produced confidence interval ensures a bot finite-sample and asymptotically optimal coverage for each $z_{new}$ almost surely.
>
> The analysis is built upon the nonparametric theory. Please let us know if you and other reviewers want more details on the analysis. We will respond timely in the following week.
>
> We note that the existing works on robust contextual optimization [7,20] also fail to achieve the individual coverage guarantee, and more important, these two existing works are restricted to certain prediction model such as k-NN or neural networks. Comparatively, our paper aims to encourage a more flexible choice of the prediction model, and the usage of uncertainty calibration method for robust optimization tasks. It is a secondary goal to achieve individual coverage guarantee, but as it shows above, the PTC framework doesn’t exclude the possibility of achieving this.
>
> Two-step procedure:
>
> Generally speaking, an ML prediction problem will become more sample efficient and easier when the conditional expectation/quantile function $E[Y|X]$ or $Q_{\tau}(Y|X)$ is smoother. In our context, the two-step procedure first predicts the conditional mean with $\hat{E}[c|z]$ and then try to predict quantiles/distribution of $c-\hat{E}[c|z]|z$. To fit the error $c-\hat{E}[c|z]|z$ can be easier than the original $c|z$ because the error’s conditional distribution is usually smoother. To see this, generally, the conditional expectation function $E[c|z]$ is highly related and very likely to wax and wane together with the quantile function $Q_{\tau}[c|z]$. In this light, subtracting $E[c|z]$ can very likely smooth out the quantile function $Q_{\tau}[c|z]$. In metaphor, this is quite like the method of controlled variates in Monte Carlo simulation which introduces a controlled random variable that is correlated with the target random variable such to reduce the variance of the estimator. Here, the conditional expectation function $E[c|z]$ works as the “controlled variate” for the original “target variate” of $Q_{\tau}[c|z]$. Another example is the nonparametric estimator presented in above. The convergence rate of the coverage guarantee scales linearly with the Lipschitz constant $L$. And the corresponding Lipschitz constant for the conditional distribution $c|z$ will be generally much larger that of the error distribution $c-\hat{f}(z)|z$.
>
> While such two-step procedure is commonly adopted in the literature of conformal prediction and ML model calibration, we provide the above explanation to justify such an approach, and we will include this discussion in the next version of our paper.
>
> Section 4:
>
> We agree that Section 4 has less coherence with other sections of the paper, because the majority of our paper studies the robust optimization problem, while this section concerns the distributionally robust optimization (DRO). Yet, they all share one theme which is to make the choice of the ML model and uncertainty calibration methods more flexible. When we reviewed the related literature, we find that the DRO papers on contextual optimization usually impose very strong assumptions to achieve the performance guarantee. To this end, while Section 4 develops no new algorithm upon the existing literature, we believe it significantly relaxes the realizability assumption and the error distribution assumptions in the existing works.
>
> We defer more responses to the raised question to Author Rebuttal. We are sorry for the inconvenience and we really appreciate your time in reading our paper and responses. Please let us know if there is any further confusion; we look forward to further discussion with you.

---

> > ### Comment · Reviewer_tVPZ · 2023-08-10
> >
> > Regarding the "PTC framework":
> >
> > While the authors occasionally use phrases such as "our predict-then-calibrate framework" within the paper, this framework is not clearly defined. As currently written, the paper presents a fairly specific problem, develops two specific methods for that specific problem, and then presents computational results for those specific methods.  For this reason, I agree with the authors that my criticisms apply to a specific methods and not of the framework, but if the authors want to invoke this as a response, then they need to provide details of their framework. It seems to me that rewriting the current paper to present a framework rather than addressing a specific problem would be creating an entirely different paper. In a paper proposing a framework, I would expect a rigorous definition of the framework, some explanation how the framework can be applied in different contexts, and some demonstration of what benefit the framework provides, such as theoretical results and/or broad experiments in a wide variety of settings where the framework might apply. I would recommend that the authors fix the concrete issues with their presented methods rather than appealing to a framework that is not clearly defined.
> >
> > Regarding the individual coverage guarantee:
> >
> > The method described in the rebuttal to produce an individual coverage guarantee makes sense. I think that the paper would be improved if this method and the associated guarantees were presented rather than the current methods.
> >
> > Regarding coverage guarantees in existing works:
> >
> > As far as I can tell, Ohmori (2021) provides no coverage guarantees of any kind, but I suspect that it should be possible to provide individualized coverage guarantees for their k-nearest-neighbors-based method, at least asymptotically, if k were scaled at an appropriate rate as the sample size increases. It is true that Chenreddy et al. (2022) provide global coverage guarantees, but the risk-averse optimization problem in Chenreddy et al. (2022) is defined differently than the one that the authors problem. In particular, the CVO problem presented by Chenreddy et al. (2022) aims to find a policy with minimal CVaR under random context, while that presented by the authors aims to find an action with minimal VaR conditioned on a context. For the former problem, the global coverage guarantee is appropriate; for the latter problem, it is not appropriate. This perhaps provides an alternative avenue for the authors to fix the issue of the incorrect coverage guarantee: instead of changing the method to provide an individual coverage guarantee, the authors could perhaps redefine the main problem to identify a policy with minimal VaR under random context.
> >
> > Regarding value of proposed work relative to existing work:
> >
> > I agree with the authors that their method appears to provide some value over existing methods by allowing additional flexibility in choice of prediction method.
> >
> > Regarding the two-step procedure:
> >
> > The authors' justification seems reasonable and I look forward to seeing the changes.
> >
> > Regarding Section 4:
> >
> > The authors claim that the method in Section 4 shares a common theme with those presented elsewhere in the paper, but this theme is not clearly communicated in the current paper, and it is difficult to see the connections. I suppose that the authors are claiming that all methods fall within the predict-then-calibrate framework, but as I mentioned earlier, this framework is not clearly defined, and the material in Sections 2,3, and 5 all deal with a specific problem rather than discussing a general framework. I still recommend removing this section unless the authors can provide a much stronger connection to the other sections than currently exists.

---

> > > ### Author Response · Authors · 2023-08-11
> > >
> > > We thank the reviewer for the prompt feedback, and we appreciate it in particular as we ourselves as reviewers haven't got to read the author's rebuttal to those papers that we reviewed :P
> > >
> > > Regarding the PTC framework:
> > >
> > > We are sorry for the caused confusion by the wording of “framework”. In fact, we intentionally avoid using the word “framework” in our paper as we don’t want to overclaim our contribution to the existing literature. The main motivation for us to write the paper is that when we see (i) the existing works on RO such as Ohmori (2021) and Chenreddy et al. (2022), while providing elegant solutions to the problem, are tied to certain ML methods; (ii) the existing works on DRO require strong assumptions such as realizability. We use the name “predict-then-calibrate” to emphasize
> > >
> > > - The two-step procedure frees out the choice of the ML model and leaves the work to uncertainty quantification methods.
> > > - The quantification of uncertainty should be made aware of downstream robust tasks, such as outputting box- or ellipsoid-shaped uncertainty sets.
> > >
> > > We used the word “framework” a bit loosely in our response (again, our apologies for that), and we didn’t mean to say our framework umbrellas all the existing methods/works. Yet, on the other hand, this disentanglement of the prediction from the calibration does give the flexibility to cover the problem of CVaR (by outputting a generative model for distributional prediction) and to extend to the case where optimality is ensured such as Gupta (2019) (by outputting ellipsoid-shaped uncertainty set).
> > >
> > > Regarding the existing works:
> > >
> > > We thank the reviewer for the detailed discussions on Ohmori (2021) and Chenreddy et al. (2022). This is indeed helpful! We agree with the comments on random context; an easier way to improve the presentation of the global coverage guarantee is to position it as a random context. For the comment on Ohmori (2021), we agree that conditional coverage can be derived in a similar manner as the roadmap in our last response for nonparametric estimators. Yet one advantage of calibrating the error distribution than the k-NN method in Ohmori (2021) is the two-step procedure argument.

---

> > > > ### Author Response · Authors · 2023-08-11
> > > > **global v.s. conditional**
> > > >
> > > > Regarding the conditional guarantee:
> > > >
> > > > We are glad that the reviewer finds our solution reasonable, and we will include this discussion in our paper. We agree that an individual/conditional guarantee would be more desirable. We’d like to add a bit more explanation for why we settled with the global/population-level guarantee in the first place. When we did the literature review for our paper, we come across the paper by Chenreddy et al. (2022), for which, we believe, the most significant contribution is to introduce the modeling philosophy of deep learning to the robust optimization community, though the paper does contain theoretical analyses. In this light, we hope our work can introduce the idea of uncertainty quantification (algorithms and analyses) to the robust optimization community, such as the scalar adjustment, the notion of global guarantee v.s. conditional guarantee, the value of the covariates, etc.
> > > >
> > > > The uncertainty quantification literature arises from two communities: the statistics community calls it conformal prediction, and the ML community calls it model calibration. For both lines of literature, the global guarantee was first studied in around the 2010s. After a decade of time, and until recent years, the conditional guarantee begins to attract more attention. As the first paper to introduce the notion of uncertainty calibration to robust optimization literature, we hope to have your understanding on that the first version of our paper aims for the global guarantee rather than the ultimate goal of conditional guarantee (of course, we are more than happy to include a few pages of detailed analyses for the proposed calibration algorithm in the last response).
> > > >
> > > > - Conformal prediction: Milestone papers [1,2] on the topic all focus on the global guarantee, and the nice survey paper [3] also focuses on the global guarantee. For conditional/individual guarantee, papers [4,5] state an “impossible triangle” for uncertainty quantification: (i) the coverage result is built for conditional coverage; (ii) it makes no assumptions on the underlying distribution; (iii) it has finite-sample guarantee rather than asymptotic consistency. Essentially, in our paper, Algorithm 1 and Algorithm 2 basically fail to meet (i) but they barely rely on any assumptions on the underlying distribution and enjoy a finite-sample guarantee, that is, achieving (ii) and (iii). The new nonparametric algorithm achieves conditional coverage, and the coverage can be finite-sample, i.e., achieving (i) and (iii). Indeed, this requires additional assumptions on the underlying distribution, which is a violation of (ii).
> > > >
> > > > - Model calibration/recalibration: [6-10] are a few state-of-the-art papers on calibrating the uncertainty of a regression model. We note that none of these papers claim to achieve a conditional coverage guarantee. Rather, they ensure a global guarantee and strive for better empirical conditional coverage. In particular, [10] gives a pessimistic result on conditional coverage and their argument is essentially based upon [4,5]; we don’t agree with its argument as it uses a very special example to establish the statement.
> > > >
> > > > As we understand, one reason this existing literature doesn’t claim achieving conditional coverage is that the Lipschitz assumption on the quantile function in our last response (though it seems to be a mild one), can’t be verified from the data a priori. We will include a full discussion on the algorithm and its analysis, but we prefer not to claim it in the first place for several reasons: (i) We don’t want to give an impression to the robust optimization community that conditional coverage is an easy task to achieve. When one applies the nonparametric algorithm (theoretically perfect), it should proceed with caution. (ii) We don’t want to give an impression to the statistics and ML communities working on uncertainty quantification problems that we underestimate the difficulty of the conditional coverage and lack of understanding of the related literature.
> > > >
> > > > We hope this better explains our preference for presenting a global guarantee in the main paper and the conditional guarantee as an extension. Again, we appreciate the time and effort spent by the reviewer and are happy to address any further concerns.
> > > >
> > > > [1] Distribution-free predictive inference for regression
> > > >
> > > > [2] Conformal prediction beyond exchangeability
> > > >
> > > > [3] A Gentle Introduction to Conformal Prediction and Distribution-Free Uncertainty Quantification
> > > >
> > > > [4] The limits of distribution-free conditional predictive inference
> > > >
> > > > [5] Conditional validity of inductive conformal predictors
> > > >
> > > > [6] Simple and scalable predictive uncertainty estimation using deep ensembles
> > > >
> > > > [7] Calibrated reliable regression using maximum mean discrepancy
> > > >
> > > > [8] Beyond pinball loss: Quantile methods for calibrated uncertainty quantification
> > > >
> > > > [9] Accurate uncertainties for deep learning using calibrated regression
> > > >
> > > > [10] Individual calibration with randomized forecasting

---

> > > > > ### Comment · Reviewer_tVPZ · 2023-08-14
> > > > >
> > > > > I appreciate the context and regarding the literature on uncertainty quantification, and the difficulties associated with providing conditional guarantees. This does help me to understand better your preference for providing a global coverage guarantee rather than a conditional one. However, it seems to me that it is more critical to provide a conditional guarantee in the setting of the contextual optimization problem defined by the authors than it is in the settings typically appearing in the uncertainty quantification literature because the conditional guarantee actually appears as a constraint in the optimization problem (specifically, the conditional coverage guarantee appears as a constraint in the reformulation provided after line 106 of the paper). As discussed in other comments, the inconsistency between the optimization problem and coverage guarantee could also be corrected by defining a different optimization problem that avoids this type of constraint.

---

> > > > > > ### Author Response · Authors · 2023-08-15
> > > > > >
> > > > > > Thanks very much for the follow-up and for spending additional time reading our response.
> > > > > >
> > > > > > We see the point on the alignment between the performance guarantee of conditional coverage and the setup of contextual (i.e., conditional) optimization, and we will include these discussions and the nonparametric calibration method above in our paper.
> > > > > >
> > > > > > Thanks again for all the helpful discussions; if there are any follow-up questions, don't hesitate to let us know, and we will respond timely in the following days.

---

> > > > ### Comment · Reviewer_tVPZ · 2023-08-14
> > > >
> > > > I appreciate the clarification.

---

### Official Review · Reviewer_4BGT · 2023-07-06

**Soundness:** 4 excellent
**Presentation:** 4 excellent
**Contribution:** 4 excellent
**Rating:** 7
**Confidence:** 3

**Summary:**

The author(s) propose a novel method of robust contextual LP, which extends the conventional contextual LP problem by allowing some uncertainties from the prediction model. It is very well-written, and states very clearly what is the problem setting, and in which direction this work advances. The model properties are discussed in detail and the implication of each theory/statement is explained in a very careful manner.


**Strengths:**

1. Clearly written and well organized
2. Provides a clear problem statement, and the algorithms are described in good detail
3.  Strong algorithm analysis with additional detials to provide context for the statements
4. Good empirical work demonstrates the method
5. The author proposes two algorithms to construct the contextual uncertainty sets $\mathcal_{U}$ (which is based on z) for the prediction model. They mention the choice of the prediction model is quite flexible, and there are a couple of advantages by doing so.


**Weaknesses:**

1. It would be nice if the author could provide the audience with some motivations for this work. For example, under what circumstances would it be beneficial to incorporate a risk-sensitive objective into contextual LP.
2. All theoretical guarantees mentioned in Section 3.1 require $\mathcal{D}_2\sim \mathcal{P}$, e.g. the validation set is correctly specified.  It would be useful to understand how misspecification impacts these results (either empirically or theoretically).


**Questions:**

1. Proposition 1 provides a coverage guarantee for Algorithm 1 and Algorithm 2. I was a bit confused by the statements. On the left hand side, the inequality goes to 0 and $|\mathcal{D}_2|$ increases. While on the right handside, the inequality goes to 1 as $|\mathcal{D}_2|$ increases. Does it mean the more observations in $\mathcal{D}_2$, the more uncertainty there will be? It would be nice if the author could provide more explanation on it.
2. As stated above, all theoretical guarantees mentioned in Section 3.1 require $\mathcal{D}_2\sim \mathcal{P}$, e.g. the validation set is correctly specified. For practical application purposes, could the author provide some strategies to make sure this requirement is guaranteed?
3. In the Experiment Section, Simple LP Visualization paragraph, the author mentions both PTC-B and PTC-E. Maybe I missed it, but it seems in Figure 2, the author only provides visualization for PTC-B. Where is PTC-E?


**Limitations:**

See above for list of areas for improvement.

---

> ### Author Rebuttal · Authors · 2023-08-10
>
> We thank the reviewer for all the comments and feedback.
>
> Typo in Proposition 1:
>
> The inequality (8) should be “>=” instead of “<=”. We thank the reviewer for noting this mistake. We agree with the intuitions mentioned by the reviewer and now the inequality becomes aligned with these intuitions.
>
> PTC-E in Figure 2:
>
> We are sorry for the caused confusion. In the setting of Figure 2, the algorithm of PTC-E will be exactly the same as PTC-B. Specifically, Figure 2 illustrates the benefits of a better prediction or calibration model in a one-dimensional case. In the one-dimensional case, the PTC-B coincides with PTC-E then because both the box-shaped and ellipsoid-shaped uncertainty sets will degenerate into line segments. We will mention the point in the next version of our paper.
>
> Validation set:
>
> That's a great point. The theoretical results hinge on the fact that the validation set comes from the same distribution as the test data. Practically, just like the standard setup of machine learning, if one has an available training dataset that is from the same distribution as the test set, then one can reserve part of the training data as the validation data. If this is not true, which is known as the out-of-domain problem, where the training data and the test data may be from different distributions, the situation doesn’t become fully pessimistic. Indeed, our algorithms essentially calibrate the uncertainty of c|z. One type of out-of-domain problem is known as the covariates shift, where the distribution of z (covariates) changes between the training data and the test data, but the conditional distribution of the target variable c|z remains the same. In this case, the theoretical guarantee of our algorithms still works. Of course, there are other setups of the out-of-domain problem that may make the algorithms fail. In our opinion, this type of setup studies the potential discrepancy between the training data and the test data, and it deserves more attention from the literature on data-driven robust optimization and contextual optimization, where it often assumes a standard i.i.d. setting. To this end, we will include more discussions about it to call for more awareness of the problem.
>
> We hope our response addresses the raised questions. We refer to our response to Reviewer 1GB8 for more discussions on the motivation of the formulation. If there are any follow-up questions/concerns, we will get back to you timely in the following discussion week.

---

### Official Review · Reviewer_VdM1 · 2023-07-06

**Soundness:** 3 good
**Presentation:** 3 good
**Contribution:** 3 good
**Rating:** 6
**Confidence:** 3

**Summary:**

The authors study a risk-sensitive contextual LP setting. They seek to predict the objective function of the LP from a context vector using a generic machine-learning algorithm, and then use this prediction to achieve a low (good) objective value in the LP. Their insight is this can be done cleverly with calibration—instead of changing the ML prediction as others have done, they calibrate the output. In theory and empirical evidence, they show their method has advantages over competitors.

**Strengths:**

- There is substantial interest in the predict-then-optimize setting for LPs (see the citations on Grigas and Elmachtoub). This setting (while not mentioned by the authors) seems to have clear applications in finance.
- The authors' work is clearly presented and claims are backed up by theory and experiments. The theory and experiments support the claims not only that the approach achieves better objective values, but gives a more detailed view of why (I particularly like 3.2).

**Weaknesses:**

- The method combines tools that we understand pretty well. In particular, it leverages the detailed picture of robust optimization linear programs that we have. The results are naturally limited to LPs.
- There isn't much discussion of limitations.

**Questions:**

1. Have the authors' investigated the application to conditional value at risk?

**Limitations:**

Not much discussion, a little bit in Future Directions.

---

> ### Author Rebuttal · Authors · 2023-08-10
>
> We thank the reviewer for bringing up the finance application. We are not quite experts in the finance domain, so we didn’t mention it in the first place. However, in the next version of our paper, we will include more discussions about it. Generally, this financial application leads to a natural consideration of CVaR objective for its convexity and time consistency.
>
> CVaR objective instead of VaR:
>
> The framework of predict-then-calibrate (PTC) is more about a generic approach that first predicts the objective vector and then quantifies the uncertainty of the prediction model for the downstream robust task. In particular, PTC specifies a procedure to characterize the conditional distribution $c|z$. For the VaR objective, it outputs box- or ellipsoid- shaped confidence set so as to meet the tractability of the robust task. In comparison, for the CVaR objective, one should replace the calibration part with a finer characterization of the conditional distribution. This indeed can be done with minor modifications of Algorithm 2 or other algorithms of a similar spirit. Note that Algorithm 2 fits a multivariate Gaussian distribution for $c|z$; with the fitted model, for newly observed covariates $z_{new}$, one can generate a number of independent samples
>
> $\tilde{c}_1,....,\tilde{c}_k$
>
> from the conditional distribution of $c|z_{new}$. Then to optimize the following empirical CVaR objective
>
> $\min_{x} \min_{\gamma} \gamma+\frac{1}{k}\sum_{j=1}^k (\tilde{c}_j x - \gamma)^+$
> $s.t. Ax\le b, x\ge 0$
>
> This is a convex program that has an equivalent linear program. We haven’t thought about this in the submitted version of our paper. However, we do find it interesting, so we will include more discussion of the CVaR objective as an extension in the next version of our paper. Moreover, we remark that the method to fit the distribution $c|z$ can be quite flexible and it can be replaced with any generative probabilistic model that outputs a distributional prediction.
>
> General optimization problem beyond LP:
>
> We presented the PTC framework under the problem of LP for that LP seems to be the most natural playground for studying robust contextual optimization. One key point here is that for the contextual-free (without $z$ or $z=1$ a.s.) case, the robust VaR formulation should have a tractable (approximate) solution; then we can utilize the structure to tailor the uncertainty calibration part. Specifically, for the robust LP problem, the context-free case follows the box- or ellipsoid-shaped uncertainty set, so the PTC algorithms inherit the structure and have the calibration model output (contextual) uncertainty set of corresponding shapes. For more general problems such as the quadratic program or convex program or even multi-stage stochastic program, as long as the context-free robust problem has a tractable solution, we can design the PTC algorithm accordingly. Compared to the context-free case, such PTC design contextualizes the uncertainty set with the context information. To this end, PTC behaves more like a plug-in module that can fit into the existing robust optimization literature. We should have included more discussions on this in our paper; we thank the reviewer for noting the point and we will include it in the next round.

---

> > ### Comment · Reviewer_VdM1 · 2023-08-16
> > **Response to authors**
> >
> > Thanks to the authors for their responses. I am generally satisfied with them (to me an other reviewers). I will keep my score.

---

### Official Review · Reviewer_1GB8 · 2023-07-07

**Soundness:** 3 good
**Presentation:** 3 good
**Contribution:** 2 fair
**Rating:** 6
**Confidence:** 3

**Summary:**

This paper considers the contextual linear optimization problem, where one is given a vector of covariates $z$ that can be used to predict a cost vector $c$, and one wishes to solve the following LP:

$ \max \ E[ c \mid  z]^T x$

$ \text{subject to}: A x = b, x \geq 0  $

This is the risk-neutral version of the contextual LP problem. The risk-sensitive version instead considers the $\alpha$-value-at-risk with respect to the conditional distribution of $c^T x$ given $z$:

$ \max \ VaR_{\alpha}( c^T x \mid z )$

$ \text{subject to}: A x = b, x \geq 0  $

The paper proposes an approximate method for solving this problem, where one approximates the VaR with a max over $c$'s in a fixed uncertainty set that depends on the given covariate vector $z$. The uncertainty set itself is obtain from a procedure that in the abstract works like this: take a prediction model $\hat{f}$ and make predictions of $c$ in a validation set. Calculate the errors/residuals, and build another model, called the calibration model, to predict these residuals. One then constructs an uncertainty set conditional on $z$ where the center comes from the prediction model and the width comes from the calibration model. This approach is shown with high confidence over the validation set to satisfy the $\alpha$ probability coverage requirement. The paper also tests this approach using synthetic toy instances as well as instances based on the contextual shortest path problem, and shows that it leads to better (lower) VaRs than existing robust-contextual proposals.

**Strengths:**

- From a novelty standpoint, I think this paper does present something new; to my knowledge the combination of value-at-risk within contextual optimization has not been done before.

- The method seems reasonably simple and could be implemented easily. It is also nice to see that there is a high-confidence guarantee on the coverage probability of the set $\mathcal{U}$ produced by Algorithms 1 and 2. Another nice aspect is that the validation data set does not need to be the same data set used to train the predictive model $\hat{f}$; this goes along nicely with the goal of robustness in the paper.

- I appreciate that the authors provided the simple examples in Section 3.2 to illustrate how the approach works and the difference between having a good prediction model vs. calibration model.

**Weaknesses:**

- I think the motivation of the paper is a little weak. While I understand the motivation for predict-then-optimize / contextual optimization, and the motivation for optimizing value-at-risk, it is not clear to me why one would want to couple these two things together. The introduction very quickly brings up the "risk-sensitive" aspect of the problem without providing an accompanying problem from the literature or from practice that would need this type of machinery.

- From a practical standpoint, there are a lot of questions that are unanswered, namely how good is the method (can one do better than this method in certain cases? see my suggestion #2 below) and how should one choose the prediction model and the calibration model?

- The numerical evaluation feels incomplete, and the method should be compared to stronger benchmarks; for this type of contextual problem, it feels like there should be lots of ways that one could represent the conditional VaR with respect to a given predictive model (see my suggestion #1 below, on using the conditional distribution outputted by the predictive model).

**Questions:**

_Question/Suggestion 1_: If the goal is to maximize the alpha-VaR conditional on a given covariate vector, could the following simpler strategy not work: in Bertsimas and Kallus (2020), the authors propose solving the contextual problem by using a machine learning model to obtain an estimate of the conditional distribution of $c$ given $z$. (For example, for a CART tree, one would run the given $z$ down the tree to determine the corresponding leaf, look at the historical $(z',c')$ pairs that are in that leaf, and consider the distribution with a weight of $1 / M$ on the $M$ pairs of $(z', c')$ values that are in that leaf, and a weight of zero on all other observations.) Bertsimas and Kallus discusses how this type of strategy can be used in conjunction with other machine learning models, such as random forests and k-nearest neighbors.

My question here is why could one not take a similar strategy here; specifically, why not output the $x$ that maximizes $\alpha$-VaR with respect to the estimated conditional distribution? So for example, for CART, this would mean minimizing $\alpha$-VaR with respect to the same discrete distribution that puts a weight of $1/M$ on the $M$ points that are mapped to the same leaf as the given $z$. (This would likely be some kind of simple MIP problem.)

In addition to the connection to the existing work of Bertsimas and Kallus, this simpler strategy is also appealing in that one does not need to specify a prediction and a calibration model separately; there is only one machine learning model used that needs to be specified.


_Question/Suggestion 2_: Is it possible to say something about how "optimal" the proposed predict-then-calibrate approach is? To elaborate, in Bertsimas, Gupta and Kallus (2017), one deals with a robust linear constraint of the form

$\mathbf{u}^T \mathbf{x} \leq b, \quad \forall \ \textbf{u} \in \mathcal{U}$

and one assumes that the actual value of $\mathbf{u}$ is the random variable $\mathbf{\tilde{u}}$. One then seeks to select $\mathcal{U}$ so that

$ \text{if} \ \mathbf{u}^T \mathbf{x} \leq b \quad \forall \mathbf{u} \in \mathcal{U}, \ \text{then} \ \mathbb{P}( \mathbf{\tilde{u}} \mathbf{x} \leq b ) \geq 1 - \epsilon$

Bertsimas, Gupta and Kallus (2018) note that this guarantee can be met if $\mathcal{U}$ is chosen as a $1 - \epsilon$-confidence region for $\mathbf{\tilde{u}}$, i.e., if $\mathbb{P}( \mathbf{\tilde{u}} \in \mathcal{U}) \geq 1 - \epsilon$, but this is not the only way that the guarantee can be met, and by choosing a smaller $\mathcal{U}$ one can still have the some probabilistic guarantee and be less conservative. Gupta (2019) shows that a number of uncertainty sets that have been proposed that are based on this confidence region idea are unnecessarily conservative, and that one can obtain much smaller uncertainty sets with the same probabilistic guarantee. The proposed methodology in the paper (replace the intractable problem on lines 106 - 107 with problem (4), and then apply either Algorithm 1 or 2) seems more aligned with this confidence region approach, so it would be great if the authors can discuss more on how tight / optimal this approach is.

_Question/Suggestion 3_: It would be nice to see better motivation for the contextual VaR formulation that the paper studies, both in the introduction and the numerics.

_Other comments/suggestions_:

Page 5, line 143: "but rather the only two options" $\to$ "but these are not the only two options".

Page 9, line 264: the word "better" is repeated here.

References:
Bertsimas, D., Gupta, V., & Kallus, N. (2018). Data-driven robust optimization. Mathematical Programming, 167, 235-292.
Bertsimas, D., & Kallus, N. (2020). From predictive to prescriptive analytics. Management Science, 66(3), 1025-1044.
Gupta, V. (2019). Near-optimal Bayesian ambiguity sets for distributionally robust optimization. Management Science, 65(9), 4242-4260.



**Limitations:**

The discussion of limitations seems adequate; as noted in my questions above, it would be nice to have some discussion of how optimal this approach is.

---

> ### Author Rebuttal · Authors · 2023-08-10
>
> We thank the reviewer for all the comments, and in particular, for the detailed suggestion improving our paper.
>
> Motivation for the robust formulation:
>
> We thank the reviewer for raising the point. Our problem setup lies at the intersection between robust optimization and contextual optimization, thus it has two positioning from the perspectives of both lines of the literature. The notion of contextual optimization has become popular in the recent decade. Our work is among the earliest efforts in working on the risk-sensitive contextual optimization problem; and compared to [7] and [20], our work emphasizes (i) the flexibility in choosing the prediction model, and (ii) the disentanglement of the prediction and the uncertainty quantification. As for the motivation, we believe it’d be better to take the perspective of robust optimization. The literature and methodology of robust optimization (RO) have been widely applied in various domains such as manufacturing, control system, energy, finance, etc. For all these RO applications, it can also be the application of the robust contextual optimization, and the framework of PTC, as long as there is a presence of covariates, which is more and more commonly available nowadays. Compared to the traditional RO methods, the PTC framework enables contextualized uncertainty set and thus provides more dynamic and less conservative solutions.
>
> Practically, robust contextual optimization has been used across many applications, such as transportation (Guo et al., 2023), portfolio management (Wang et al. 2022), and healthcare (Gupta, Vishal, et al. 2020). Also with a few more applications mentioned in [7,23]. We will include more discussions on motivation in the next version of our paper.
>
> Hardness of the problem and MIP formulation:
>
> The robust optimization, even without the covariates, is generally NP-hard due to joint optimization over the decision variable and the uncertainty set. We adopt an approximation approach that gives up the optimization over the uncertainty set but just sticks to one single uncertainty set. The mixed integer approach in Bertsimas and Kallus (2020) takes another route of reformulating the problem as a mixed integer program (MIP). The MIP reformulation doesn’t change the hardness of the problem but is an exact reformulation, and for small-scale problems, it offers an exact solution. We should have mentioned this in our paper, and we appreciate the reviewer for pointing it out.
>
> Compatibility with other existing results:
>
> In the paper, we mainly present the PTC framework for approximately solving the robust optimization (RO) problem, and generally, such a route lacks a theoretical guarantee on its optimality (even for the context-free case). Meanwhile, the PTC framework is also compatible with the MIP reformation, and when there is an optimality guarantee for the context-free robust optimization, the PTC framework is capable of migrating the result to the contextual case.
>
> - One-step approach v.s. two-step approach: Both approaches aim for predicting the distribution of $c|z.$ The literature on uncertainty quantification/conformal prediction usually adopts this two-step approach, and this in fact has a theoretical justification. The two-step procedure first predicts the conditional mean with $\hat{E}[c|z]$ and then tries to predict quantiles/distribution of $c-\hat{E}[c|z]|z$. To fit the error $c-\hat{E}[c|z]|z$ can be easier than the original $c|z$ because the error conditional distribution is usually smoother, and ML models can be more sample efficient when fitting smoother functions. To see this, generally, the conditional expectation function $E[c|z]$ is highly related and very likely to wax and wane together with the quantile function $Q_{\tau}[c|z]$. In this light, subtracting $E[c|z]$ can very likely smooth out the quantile function $Q_{\tau}[c|z]$. In metaphor, this is quite like the method of controlled variates in Monte Carlo simulation which introduces a controlled random variable that is correlated with the target random variable to reduce the variance of the estimator. Here, the conditional expectation function $E[c|z]$ works as the “controlled variate” for the original “target variate” of $Q_{\tau}[c|z]$. Between these two approaches, we don’t argue for the superiority of one over the other, and one can follow the training-validation approach to pick the better one for a specific dataset.
>
> - Our algorithms are compatible with the MIP formulation of the RO problem. For example, once we obtain the output of Algorithm 2, we can use it as a generative model to generate samples from the conditional distribution of $c|z$. And then we can plug these samples in the MIP formulation in Bertsimas and Kallus (2020).
>
> - We thank the reviewer for bringing our attention to Gupta (2019). We weren’t aware of the work and read it with great interests. The uncertainty set constructed by Gupta (2019) has an ellipsoid form. For the contextual setting, the PTC framework can also predict the conditional mean and conditional variance of the posterior distribution as the parameters for the ellipsoid for some fixed prior distribution. Then, one can construct the uncertainty set in a similar manner as Gupta (2019). Consequently, under some additional assumptions, one can also prove similar optimality results as in Gupta (2019) for the contextual LP problem. We will include more discussions on this in the next version of our paper.
>
> We also thank the reviewers for pointing out some typos in our writing. In the coming week, we will get back to you timely for follow-up questions.

---

> > ### Comment · Reviewer_1GB8 · 2023-08-11
> >
> > Thank you for these clarifications.

---

### Official Review · Reviewer_kcWq · 2023-07-26

**Soundness:** 2 fair
**Presentation:** 3 good
**Contribution:** 3 good
**Rating:** 6
**Confidence:** 3

**Summary:**

This paper considers a risk-sensitive version of the contextual LP problem by replacing the original risk-neutral expected cost objective with VaR. The authors propose a new paradiam termed "predict-then-calibrate" that first learns a prediction model, and then uses calibration to quantify the uncertainty of the prediction. They present two algorithms that output a box/ellipsoid uncertainty set. The authors then provide a coverage guarantee for both algorithms, and conduct empirical experiments that show favorable properties of the proposed algorithms.

**Strengths:**

- This paper considers contextual linear optimization problems with a risk-sensitive objective, which is a good complement to existing risk-neutral contextual linear optimization literature. It also complements the conditional robust optimization literature by providing a flexible algorithm in terms of the modeling choices. The idea of adding a calibration step in contextual linear optimization is novel.
- The two algorithms provided are intuitive and flexible to implement.
- The paper is overall clearly written and well-structured. The authors use simple intuitive examples to illustrate the value of better prediction and calibration, which I find really helpful.

**Weaknesses:**

- The modeling choice for $\hat{h}$ seems vague to me. More elaborations on this subject would be helpful.
- I have some concerns regarding the comparisons in the empirical session. See "questions" for details.

**Questions:**

- Can the authors provide more guidelines for the modeling choice of $\hat{h}$? In practice, how to determine which model is better for $\hat{h}$?
- In Figure 4 and Table 1, the authors use the Kernel Ridge method with the RBF kernel as the prediction model and the NN model as the preliminary calibration model in both PTC-B and PTC-E (I found this in the appendix; please state it in the main text). Since RBF kernel works the best in Figure 3, this looks like a somewhat unfair comparison, since we are comparing the best among PTC-B/PTC-E with other algorithms. Can the authors elaborate more on this? Also, how does the run time of difference algorithms compare?
- Is it possible to empirically demonstrate how good the proposed algorithms do in terms of individual-level coverage?


**Limitations:**

The authors discussed the limitations in the conclusion session.

---

> ### Author Rebuttal · Authors · 2023-08-10
>
> We thank the reviewer for all the comments and feedback, and we hope our response to the raised questions further clarifies the positioning of the predict-then-calibrate framework, and in what way it is connected with the existing results and potential future works.
>
> The choice of model $\hat{h}:$
>
> In short, the vector function $\hat{h}$ specifies the shape of the confidence set. In other words, the vector function $\hat{h}$ captures the heteroscedasticity (with respect to the covariates $z$) of the prediction residual. Essentially, the confidence interval reduces to the quantile prediction, which results in the choice of pinball loss here. The optimization/learning problem of (5) aims to fit a quantile model to the residuals $r_{ti}$’s. So $\hat{h}$ can be chosen as any model that is compatible with the optimization of the pinball loss, including linear models, NNs, and gradient boosting regression. A natural criterion for selecting $\hat{h}$ would be to select the one with the smallest empirical loss of (5) as this will lead to the most accurate quantile prediction (and hence the confidence set prediction).
>
> More discussions on Figure 4 and Table 1:
>
> The reviewer raised a nice point that the implementation of PTC-B/PTC-E uses several different prediction models such as linear models, RBF kernel, and NN. Meanwhile, for the benchmark methods, the method “kNN” is indeed tied to the use of the k-NN as the prediction model, and the methods “DCC” and “IDCC” are tied to the use of the neural network model. They can’t be adapted to other prediction models. This in fact highlights that in handling robust contextual optimization problems, the predict-then-calibrate (PTC) proceeds in two steps, first predict and then quantify the prediction uncertainty, while these existing methods are a one-step procedure. The two-step procedure offers flexibility in choosing the prediction model and thus can best utilize the power of the off-the-shelf ML toolboxes. To some extent, we think this PTC framework is not quite about one new method to outperform the existing benchmarks, but to extend these existing methods to a larger scope -- first, develop the best ML prediction model, and then properly characterize the prediction uncertainty. Importantly and ideally, such disentanglement of prediction and uncertainty quantification provides better empirical performance and theoretical tractability.
>
> Run time:
>
> For the uncertainty quantification part, the run time of PTC-B and PTC-E algorithms depends on the selected models in prediction and calibration.
>  For the downstream robust optimization problem, the runtime depends on the shape of the specified uncertainty set. There are three types/shapes of uncertainty set in the experiments. (1) Box-shaped uncertainty sets for the PTC-B algorithm, which leads to an LP by reformulating the RO problem, so it is the most computationally efficient. (2) Ellipsoid-shaped uncertainty sets for the kNN and PTC-E algorithms in comparison, which leads to a second-order conic programming problem in the reformulation of the RO problem. (3) Specific-shaped uncertainty sets for the DCC and IDCC algorithms proposed in (Chenreddy et al. 2022), where a decomposition-based solving method has been presented in their paper and the termination criterion is defined by the tolerance gap or maximum iteration, so the run time also depends on the self-defined settings. Generally speaking, the run time is (1)<(2)<(3). Considering that the run time of (1) and (2) is not much different in our experiment, while the run time of (3) depends on their own Settings, and the time is not long (each instance is basically no more than half a minute), so we didn’t compare it in the paper.
>
> Individual coverage:
>
> We mentioned the limitation of global coverage in our paper. In fact, this is not an intrinsic problem associated with the PTC framework. If one aims for individual coverage, one can choose accordingly an uncertainty quantification method that satisfies such an objective. We invite the reviewer to check our response to Reviewer tVPZ for more details if you are interested and have time.
>
> We hope the above response addresses the raised questions, and if there are any further questions/confusions, we will respond timely in the following discussion week.

---

### Author Rebuttal · Authors · 2023-08-10

We thank the reviewers for spending the time reading our paper, and for all the helpful comments. The raised questions inspire us to think about important aspects that we haven't come across when we write the paper. We look forward to further discussions in the coming week.

We'd like to take the extra space here to further address a few comments made by Reviewer tVPZ. Our apologies for the confusion and the caused inconvenience.

Scalar adjustment:

One very simple adjustment that appears in both of our algorithms is the scalar adjustment, i.e., the choice of $\eta$. It is a very simple yet effective way to ensure the guarantee. As far as we know, this (surprisingly) seems to be the first such design in robust optimization, while many other works need a complicated procedure to ensure the coverage guarantee.

Motivation for our algorithm design:

As noted above and in our paper, our goal is to advocate for a more flexible choice of the prediction model and the uncertainty calibration method and also to show the value of contextual information in robust optimization. The framework is compatible with many algorithm designs such as the above nonparametric design, our response to reviewer VdM1 on CVaR objective, and our response to reviewer 1GB8 on mixed integer reformulation. We choose Algorithm 1 and Algorithm 2 as exemplifiers for the PTC framework because they naturally inherit the box- and ellipsoid-shaped uncertainty sets in context-free robust optimization. Algorithm 1 fits an ML model to predict the side length of the box while Algorithm 2 predicts the shape of the ellipsoid. We agree with the reviewer that the theoretical guarantees in Proposition 1 and Corollary 1 do not exclude the possibility of having a bad uncertainty set. First, we don’t observe such a phenomenon in our numerical experiments. Second, the PTC framework is generally compatible with other algorithm designs such as parametrized uncertainty sets (Wang et al. 2023, Learning for Robust Optimization).

Optimality ratio:

The robust optimization problem, even for the context-free case, is an NP-hard problem in general due to the joint optimization over the decision variable and the uncertainty set. We didn’t provide any optimality ratio guarantee in our paper; neither does general robust optimization literature. In other words, the lack of optimality ratio guarantee is not caused by the contextual setup, the PTC framework/algorithms, or our analysis; it is determined by the nature of the robust optimization problem. However, we don’t exclude the possibility of achieving an optimality guarantee or a finer analysis. For example, as our response to reviewer 1GB8, we believe that under the same conditions as Gupta (2019), a similar optimality ratio guarantee can be achieved under the PTC framework and also as per our examples and discussions in Section 3.2. We thank the reviewer for raising the point and will include more discussion on this.

Again, we appreciate the time spent by the reviewers reading our paper and all the comments. We look forward to further discussions in the coming week.

---

### Decision · Program_Chairs · 2023-09-21

**Decision:**

Accept (poster)

**Comment:**

The papers consider risk-averse objectives for contextual linear optimization. The reviewers appreciated the paper but raised some concerns. Overall the paper studies a new formulation of an important problem an interesting way and would be valuable addition to the conference. The authors should address the reviewers concerns as outlined in the rebuttal in the text of the paper when preparing a camera ready.